# OVERCOMING BARRIERS TO THE TRAINING OF EFFECTIVE LEARNED OPTIMIZERS

## ABSTRACT

In this work we focus on general-purpose learned optimizers capable of training a wide variety of problems with no user-specified hyperparameters. We introduce a new, neural network parameterized, hierarchical optimizer with access to additional features such as validation loss to enable automatic regularization. Most learned optimizers have been trained on only a single task, or a small number of tasks. We train our optimizers on thousands of tasks, making use of orders of magnitude more compute, resulting in optimizers that generalize better to unseen tasks. The learned optimizers not only perform well, but learn behaviors that are distinct from existing first order optimizers. For instance, they generate update steps that have implicit regularization and adapt as the problem hyperparameters (e.g. batch size) or architecture (e.g. neural network width) change. Finally, these learned optimizers show evidence of being useful for out of distribution tasks such as training themselves from scratch.

## 1 INTRODUCTION

Much of the success of modern deep learning has been driven by a shift from hand-designed features carefully curated by human experts, to domain-agnostic methods that can learn features from large amounts of data. By leveraging large-scale datasets with flexible models, we are now able to rapidly learn powerful features for new problem settings that often generalize to novel tasks. While learned features outperform hand-designed features on numerous tasks (Krizhevsky et al., 2012; Berner et al., 2019; Vinyals et al., 2019; Piech et al., 2015), we continue to use hand-designed optimization algorithms (such as gradient descent, momentum, and so on) for training models.

These hand-designed update rules benefit from decades of optimization research but still require extensive expert supervision in order to be used effectively in machine learning. For example, they fail to flexibly adapt to new problem settings and require careful tuning of learning rate schedules and momentum timescales for different model architectures and datasets (Choi et al., 2019). In addition, most do not leverage alternative sources of information beyond the gradient, such as the validation loss. By separating the optimization target (training loss) from the broader goal (generalization), classic methods require more careful tuning of regularization and/or data augmentation strategies by the practitioner.

To address these drawbacks, recent work on *learned* optimizers aims to replace hand-designed optimizers with a parametric optimizer, trained on a set of tasks, that can then be applied more generally. Recent work in this area has focused on either augmenting existing optimizers to adapt their own hyperparameters (Daniel et al., 2016; Xu et al., 2017; 2019), or developing more expressive learned optimizers to replace existing optimizers entirely (Andrychowicz et al., 2016; Wichrowska et al., 2017; Lv et al., 2017; Metz et al., 2018; 2019a;b; Gu et al., 2019). These latter models take in problem information (such as the current gradient of the training loss) and iteratively update parameters. However, to date, learned optimizers have proven to be brittle and ineffective at generalizing across diverse sets of problems.

Our work identifies fundamental barriers that have limited progress in learned optimizer research and addresses several of these barriers to train effective optimizers:

1. **Computational scale**: Training a learned optimizer is costly. When training the optimizer, a single training step requires applying the optimizer to a training task for some number

of unrolled steps. This work utilizes massive parallel computing infrastructure to scale the number of unrolled steps an order of magnitude larger than in previous work.

2. **Training tasks**: Deep learning requires large training datasets. For learned optimizers to be effective, we similarly need a large dataset of *optimization tasks* on which to train. We build off of the TaskSet dataset (Metz et al., 2020) and construct a dataset of more than a six thousand diverse optimization tasks commonly found in machine learning. We show how this large and diverse task distribution is critical for training optimizers that generalize.

3. **Inductive bias of optimizer architecture**: The parameterization of the learned optimizer and the task information fed to it both strongly affect performance. We propose a new hierarchical learned optimizer architecture that incorporates additional task information (such as validation loss), and show that it outperforms previous learned optimizer architectures.

By addressing these barriers, we develop learned optimizers that exceed prior work in scale, robustness, and out of distribution generalization. As a final test, we show that the learned optimizer can be used to train new learned optimizers from scratch (analogous to "self-hosting" compilers (Hart and Levin, 1962)).

## 2 PRELIMINARIES

Training a learned optimizer is a bilevel optimization problem that contains two loops: an *inner* loop that applies the optimizer to solve a task, and an *outer* loop that iteratively updates the parameters of the learned optimizer (Franceschi et al., 2018). We use the *inner-* and *outer-* prefixes throughout to be explicit about which optimization loop we are referring to. That is, the *inner-loss* refers to a target task's loss function that we wish to optimize, and the *outer-loss* refers to a measure of the optimizer's performance training the target task (inner-task). Correspondingly, we refer to the optimizer parameters as *outer-parameters*, and the parameters that the optimizer is updating as *inner-parameters*. *Outer-optimization* refers to the act of finding outer-parameters that perform well under some *outer-loss*.

For a given inner-task, we apply the learned optimizer for some number of steps (*unrolling* the optimizer). Ideally, we would unroll each target task until some stopping criterion is reached, but this is computationally infeasible for even moderate scale machine learning tasks. *Each* outer-iteration requires unrolling the optimizer on a target task. Short (truncated) unrolls are more computationally efficient, but suffer from truncation bias Wu et al. (2016); Metz et al. (2019b) in that the outer-loss surface computed using truncated unrolls is different (and may have different minima) than the fully unrolled outer-loss.

## 3 METHODS: ADDRESSING THE THREE BARRIERS TO LEARNED OPTIMIZERS

### 3.1 OUTER-OPTIMIZATION

To train the optimizer, we minimize an outer-loss that quantifies the performance of the optimizer. This is defined as the mean of the inner-loss computed on the inner-*validation* set for some number of unrolled steps, averaged over inner-tasks in the outer-training taskset. Although this outer-loss is differentiable, it is costly to compute the outer-gradient (which involves backpropagating through the unrolled optimization). In addition, the outer-loss surface is badly conditioned and extremely non-smooth Metz et al. (2019b), making it difficult to optimize.

We deal with these issues by using derivative-free optimization–specifically, evolutionary strategies (ES) Rechenberg (1973)–to minimize the outer-loss, obviating the need to compute derivatives through the unrolled optimization process. Previous work has used unrolled derivatives (Andrychowicz et al., 2016; Wichrowska et al., 2017; Metz et al., 2019b), and was thus limited to short numbers of unrolled steps (e.g. 20 in Andrychowicz et al. (2016) and starting at 50 in Metz et al. (2019b)). Without ES, the gradient estimates we obtain are extreamly high variance to the point that no training occurs. Using ES, we are able to use considerably longer unrolls. Initial unroll lengths were chosen to balance communication cost between parallel workers (updating optimizer parameters) with the computational cost of unrolling on individual workers (estimating the local gradient with ES). We start outer-training by sampling unroll steps uniformly from 240-360. When performance saturates with

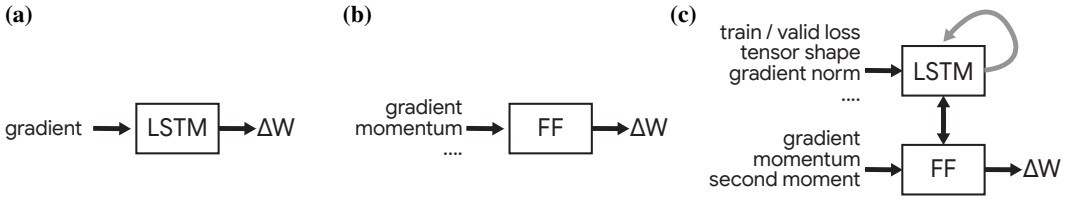

Figure 1: **(a)** The learned optimizer architecture proposed in Andrychowicz et al. (2016) consisting of a per-parameter LSTM. **(b)** The learned optimizer architecture proposed in Metz et al. (2019b) consisting of a per-parameter fully connected (feed-forward, FF) neural network with additional input features. **(c)** The learned optimizer architecture proposed in this work consisting of a per-tensor LSTM which exchanges information with a per-parameter feed forward neural network (FF). The LSTMs associated with each tensor additionally share information with each other (shown in gray).

these biased gradients, continue training with an unbiased, but more expensive, gradient estimator (see Appendix B).

ES and PES have an additional benefit, in that optimizing with ES smooths the underlying loss function. This smoothing helps stabilize outer-training (Metz et al., 2019b). We set the standard deviation of the Gaussian distribution used by the ES algorithm (which also controls how much the outer-loss is smoothed) to 0.01. To deal with the high variance of the ES estimate of the gradient, we use antithetic sampling and train in in parallel using 1024 multi-core CPU workers. While using more workers increases training speed, we find 1024 to be the point where performance gains become sub-linear. For more details see Appendix B.

### 3.2 Task distributions

To train the optimizer, we need to define a set of inner-tasks to use for training. The choice of training tasks is critically important for two reasons: it determines the ability of the optimizer to outer-generalize (i.e. the learned optimizer's performance on new tasks), and it determines the computational complexity of outer-training. For improved outer-generalization, we would like our inner-problems to be representative of tasks we care about. In this work, these are state-of-the-art machine learning models such as ResNets (He et al., 2016a) or Transformers (Vaswani et al., 2017). Unfortunately, directly utilizing these large scale models is computationally infeasible, therefore we outer-train on proxy tasks for speed(Zoph et al., 2018).

In order to outer-train a learned optimizer capable of generalizing to new optimization tasks, we utilize an outer-training task set consisting of around 6,000 tasks designed after Metz et al. (2020). These tasks include RNNs (Hochreiter and Schmidhuber, 1997; Chung et al., 2014), CNNs (LeCun, 1998), masked auto regressive flows (Papamakarios et al., 2017), fully connected networks, language modeling, variational autoencoders (Kingma and Welling, 2013), simple 2D test functions, quadratic bowls, and more. For tasks that require them, we additionally sample a dataset, batch size, network architecture, and initialization scheme. To keep outer-training efficient, we ensure that all tasks take less than 100 milliseconds per-training step. For each task that makes use of a dataset, we create four splits of the data to prevent leakage: training data, which we compute gradients on and use to update the inner-parameters; inner-validation data, which is used to compute validation losses used by the learned optimizer; outer-validation data, which is used to update the weights of the learned optimizer; and test data, which is used to test an already trained learned optimizer. Because loss values vary in magnitude, when outer-training we normalize these outer-loss values by the best loss achieved by a baseline optimizer and the initial loss value. Note this normalization is not used during inner-training.

### 3.3 Optimizer architecture

Designing a learned optimizer architecture requires balancing computational efficiency and expressivity. Past work in learned optimizers has shown that incorporating inductive biases based on existing optimization techniques such as momentum or second moment accumulation leads to better performance (Wichrowska et al., 2017; Metz et al., 2019b). The optimizer we use in this work consists of a hierarchical optimizer similar to (Wichrowska et al., 2017) (Figure 1). A per-tensor LSTM is run on features computed over each parameter tensor. This LSTM then forwards information to the

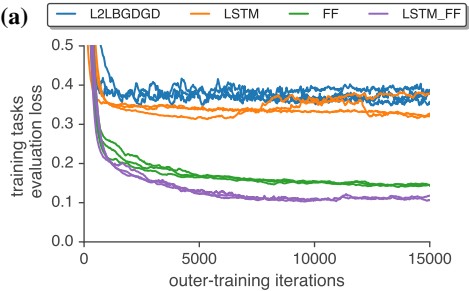 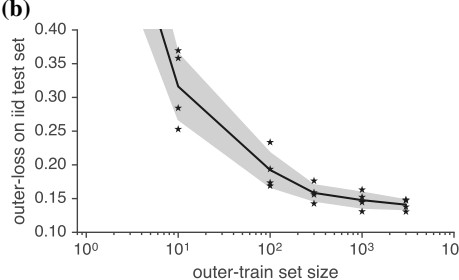

Figure 2: **(a)** Our proposed learned optimizer has a greater sample efficiency than existing methods. On the x-axis we show outer-updates performed to outer-train each learned optimizer. On the y-axis we show outer-loss. Each point consists of using a single outer parameter value to train 100 models for 10k inner-iterations, each with five random initializations. We show the average validation performance post-normalization averaged over all tasks, seeds, and inner-training steps. **(b)** Outer-generalization performance increases with an increasing number of outer-training tasks. We show the mean performance achieved after ∼10000 outer-training iterations (between 10500 and 9500) averaged over four random seeds while varying outer training set size. We show each seed as a star, and standard deviation as the shaded region. We find poor outer-generalization for smaller numbers of tasks. As we increase the number of outer-training tasks performance becomes more consistent, and lower loss values are achieved.

other tensors' LSTMs as well as to a per-parameter feedforward neural network. The per-parameter feedforward network additionally takes in information about gradients and parameter value, and outputs parameter updates. Additional outputs are aggregated and fed back into the per-tensor network. This information routing allows for communication across all components.

For per-parameter features we leverage effective inductive biases from hand-designed optimizers, and use a variety of features including the gradient, the parameter value, and momentum-like running averages of both. All features are normalized, in a fashion similar to that in RMSProp (Tieleman and Hinton, 2012), or relative to the norm across the full tensor. For per-tensor features we use a variety of features built from summary statistics computed from the current tensor, the tensor's gradient, and running average features such as momentum and second moments. We also include information about the tensor's rank and shape. We also feed global features into the per-tensor LSTM, such as training loss and validation loss, normalized so as to have a relatively consistent scale across tasks. To compute a weight update, the per-parameter MLP outputs two values, $(a, b)$, which are used to update inner-parameters: $w^{t+1} = w^t + \exp(a)b$. See Appendix C for many more details.

# 4 RESULTS

## 4.1 COMPARING LEARNED OPTIMIZER ARCHITECTURES AND TRAINING TASK SET SIZES

First, we show experiments comparing the performance of different learned optimizer architectures from the literature. We trained: a component-wise LSTM optimizer from Andrychowicz et al. (2016) (L2LBGDGD), a modification of this LSTM with the decomposed direction and magnitude output from Metz et al. (2019b) (LSTM), the fully connected optimizer from Metz et al. (2019b) (FF), as well as the proposed learned optimizer in this work (§3.3) (LSTM_FF). As shown in Figure 2(a), the proposed architecture achieves the lowest outer-training loss and achieves this in the fewest outer-training steps. To the best of our knowledge, this is the first published comparison across *different* learned optimizer architectures, on the same suite of tasks. Previous work only compared a proposed learned optimizer against hand-designed (baseline) optimizers.

Next, we explored how increasing the number of inner-tasks used when training an optimizer affects final performance. To do this, we randomly sampled subsets of tasks from the full task set, while evaluating performance on a common held-out set of tasks. Figure 2(b) shows that increasing the number of tasks leads to large improvements.

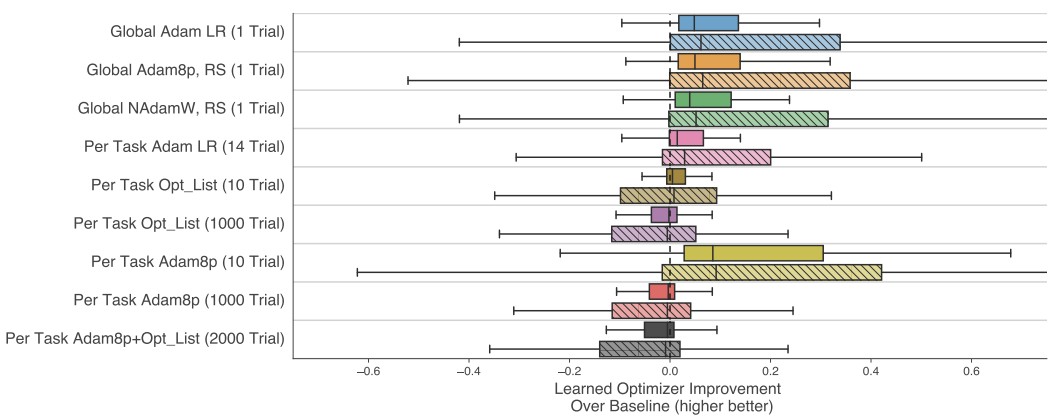

Figure 3: Learned optimizer performance as compared to a battery of different baselines. For each baseline, we show a box plot over 100 different tasks. We show both a training set (solid color) which the optimizer has seen, and a test set (hatched light color) which is sampled from the same distribution but never used for outer-training. Values to the right of the dashed line indicate the learned optimizer outperforming the corresponding baseline. The learned optimizer outperforms any single baseline optimizer with a fixed hyperparameter configuration (marked global, with a single Trial). We also outperform per-task tuning of baselines, when tuning is done only over a modest number of hyperparameter configurations (for instance, learning rate tuned Adam which uses 14 trials). Note that the learned optimizer undergoes no per-task tuning and only makes use of one trial.

## 4.2 COMPARISONS WITH HAND-DESIGNED OPTIMIZERS

We compare against three baseline optimizers: *AdamLR*, which is the Adam optimizer Kingma and Ba (2014) with a tuned learning rate. *Adam8p*, which is a version of the Adam optimizer with eight tunable hyperparameters: learning rate, $\beta_1$, $\beta_2$, and $\epsilon$, plus additional $\ell_1$ and $\ell_2$ regularization penalties, and a learning rate schedule parameterized with a linear decay and exponential decay. See Appendix D for more details. Our final baseline optimizer, called *opt_list*, consists of the NAdam optimizer (Dozat, 2016) with "AdamW" style weight decay (Loshchilov and Hutter, 2017), cosine learning rate schedules (Loshchilov and Hutter, 2016) and learning rate warm up (See Metz et al. (2020) for more info). Instead of tuning these with some search procedure, however, we draw them from a sorted list of hyperparameters provided by (Metz et al., 2020) for increased sample efficiency.

Evaluation of optimizers, let alone learned optimizers, is difficult due to different tasks of interest, hyperparameter search strategies, and compute budgets (Choi et al., 2019; Sivaprasad et al., 2019). We structure our comparison by exploring two scenarios for how a machine learning practitioner might go about tuning the hyperparameters of an optimizer. First, we consider an "off-the-shelf" limit, where a practitioner performs a small number of optimizer evaluations using off-the-shelf methods (for instance, tuning learning rate only). This is typically done during exploration of a new model or dataset. Second, we consider a "finely tuned" limit, where a practitioner has a large compute budget devoted to tuning hyperparameters of a traditional optimizer for a particular problem of interest.

For the "off-the-shelf" limit, we consider the scenario where a practitioner has access to a limited number of optimization runs (trials) ($\leq 10$) for a particular problem. Thus, we select a first set of baseline hyperparameters using a single, default value (denoted *global* in Fig 3) across all of the tasks. We use random search (RS) using 1000 different hyperparameter values to find the global value that performs best on average for all tasks. Practitioners often tune models with a small number of evaluations. As such, we include comparisons to per-task tuned learning rate tuned Adam, the first 10 entries of opt_list, and 10 hyperparameter evaluations obtained from random search using the adam8p hyperparmeterization.

For the "finely tuned" limit, we consider task-specific hyperparameter tuning, where the hyperparameters for each baseline optimizer are selected individually for each task (*per-task* in Fig 3).

We plot a histogram over tasks showing the difference in performance between the each baseline optimizer and the learned optimizer in each row of Fig 3. First, we note that the distribution is broad, indicating that for some tasks the learned optimizer is much better, whereas for others, the baseline

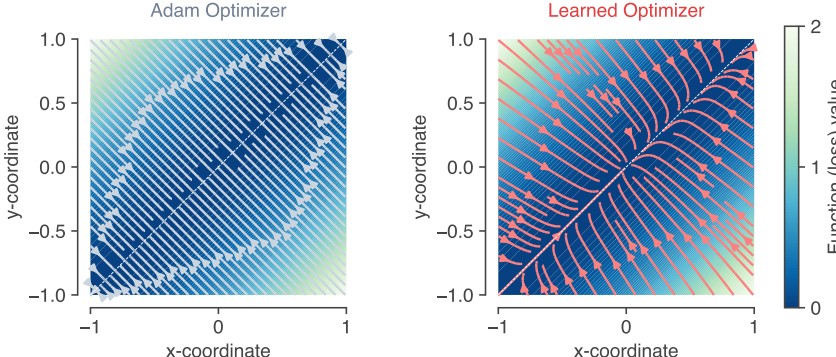

Figure 4: Implicit regularization in the learned optimizer. Panels show optimization trajectories over a 2D loss surface, $f(x, y) = \frac{1}{2}(x - y)^2$, that has a continuum of solutions along the diagonal. **Left:** Trajectories of the Adam optimizer go to the nearest point on the diagonal. **Right:** Trajectories of the learned optimizer go towards the diagonal, but also decay towards a single solution at the origin, *as if* there were an added regularization penalty on the magnitude of each parameter.

optimizer(s) are better. On average, we see small but consistent performance improvements over baseline optimizers, especially in the "off-the-shelf" scenario. We attribute this to the diverse set of tasks used for training the learned optimizer.

### 4.3 UNDERSTANDING OPTIMIZER BEHAVIOR

To better understand the behavior of the learned optimizer, we performed probe experiments where we compared trajectories of the learned optimizer on simple loss surfaces against baseline optimizers. The goal of these experiments was to generate insight into *what* the learned optimizer has learned.

In machine learning, many problems benefit from including some kind of regularization penalty, such as an $\ell_2$ penalty on the weights or parameters of a model. We explored whether the learned optimizer (which was trained to minimize validation loss) had any *implicit* regularization, beyond what was specified as part of the loss. To test this, we ran optimizers on a simple 2D loss surface, with a continuum of solutions along the diagonal: $f(x, y) = \frac{1}{2}(x - y)^2$. Although any point along the $x = y$ diagonal is a global minimum, we wanted to see if the learned optimizer would prefer any particular solution within that set. Figure 4 shows the resulting training trajectories along the 2D loss surface from many starting points. For a baseline optimizer (left), the trajectories find the nearest point on the diagonal. However, we find that the learned optimizer has learned some implicit regularization, in that it pushes the parameters towards a solution with small norm: $(0, 0)$.

### 4.4 GENERALIZATION ALONG DIFFERENT TASK AXES

Next, we wondered whether the learned optimizer was capable of training machine learning models which differed across different architectural and training hyperparameters. To test this type of generalization, we trained fully connected neural networks on CIFAR-10 and MNIST, and swept three model hyperparameters: the number of hidden units per layer (network width), the batch size, and the number of training examples (dataset size, formed by subsampling the full dataset). For each sweep, we compare the learned optimizer to a baseline optimizer, Adam, over a grid of eight different learning rates logarithmically spaced from $10^{-5.5}$ to $10^{-2}$.

The results of these experiments are in Fig. 5. As we vary the number of hidden units (left column) or batch size (middle column), the learned optimizer generalizes outside of the range of hidden units used during training (indicated by the shaded regions). In addition, the learned optimizer matches the performance of the best learning rate tuned Adam optimizer. On CIFAR-10, as we move further away from the outer-training task distribution, the learned optimizer diverges. For dataset size (right column), we find that the learned optimizer is more sensitive to the amount of data present (performance drops off more quickly as the dataset size decreases). These experiments demonstrate the learned optimizer is capable of adapting to some aspects of target tasks which differ from its outer-training distribution, without additional tuning.

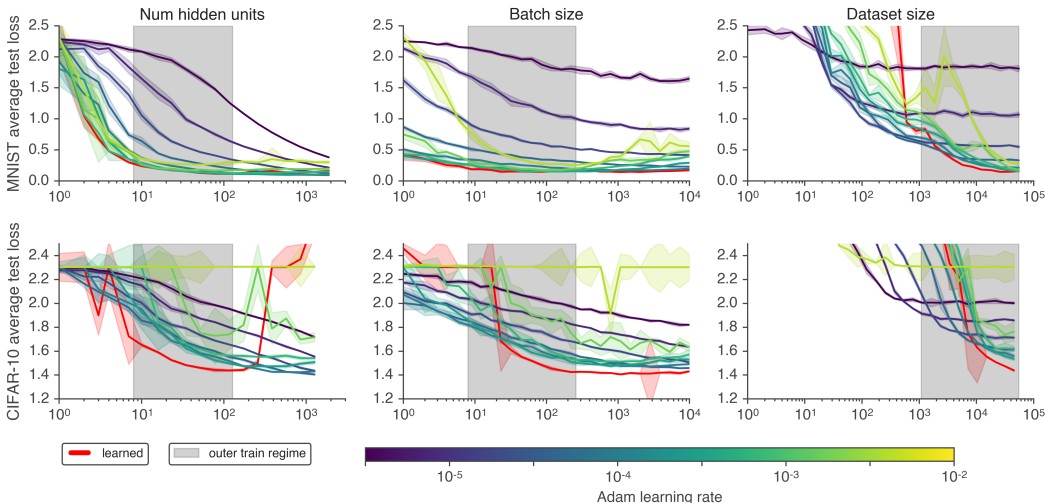

Figure 5: We show outer-generalization of the learned optimizer in a controlled setting by varying hyperparameters of the task being trained. We vary parameters around two types of models. **Top row:** A two hidden layer, 32 unit fully connected network trained on MNIST with batch size 128. **Bottom row:** A two layer hidden layer, 64 unit fully connected network trained on CIFAR-10 with batch size 128. We vary the width of the underlying network, batch size, and dataset size in each column. Each point on each curve shows the mean test loss averaged over 10k inner steps. We show median performance over five random task initializations. Error bars denote one standard deviation away from the median. Color denotes different learning rates for Adam log spaced every half order of magnitude for Adam with purple representing $10^{-6}$, and yellow at $10^{-2}$. We find the learned optimizer is able to generalize outside of the outer-training distribution (indicated by the shaded patch) in some cases.

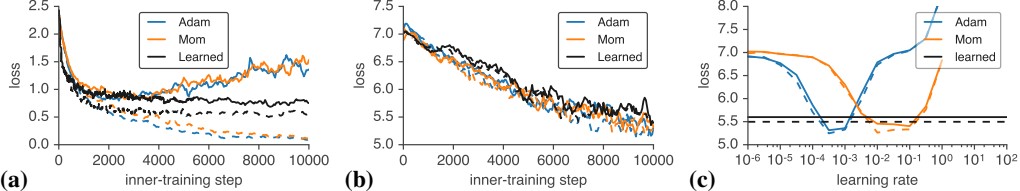

Figure 6: Learned optimizers are able to optimize ResNet models. **(a)** Inner-learning curves using the learned optimizer to train a small ResNet on CIFAR-10. We compare to learning rate tuned Adam and Momentum. Solid lines denote test performance, dashed lines are train. **(b)** Inner-learning curves using the learned optimizer to train a small ResNet on 64x64 resized ImageNet. We compare to learning rate tuned Adam and Momentum. **(c)** Performance averaged between 9.5k and 10k inner-iterations for Adam and Momentum as a function of learning rate, on the same task as in (b). Despite requiring no tuning, the learned optimizer performs similarly to these baselines after tuning.

## 4.5 GENERALIZATION TO LARGE-SCALE PROBLEMS

We test the learned optimizer on large-scale machine learning tasks. We use two ResNet V2 (He et al., 2016b) architectures: a 14 layer residual network trained on CIFAR-10 (Krizhevsky et al., 2009); and a 35 layer residual network trained on 64x64 resized ImageNet (Russakovsky et al., 2015). We train both networks with the learned optimizer and compare the performance to learning rate tuned Adam and Momentum (Fig. 6). Details in Appendix D. For CIFAR-10, we find that the learned optimizer achieves similar performance as the baselines but does not overfit later in inner-training. For ImageNet, we find that the learned optimizer performs slightly worse. We expect modifying the underlying outer-training task distribution to include more problems similar to this evaluation problem will improve performance.

Note that our baselines only include learning rate tuning. More specialized hyperparameter configurations, designed specifically for these tasks, such as learning rate schedules and data augmentation

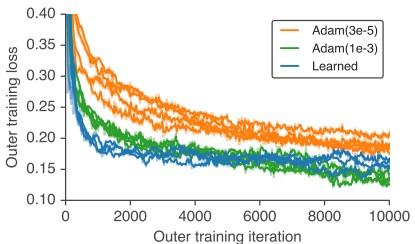

Figure 7: The learned optimizers can be used to train themselves about as efficiently as hand designed methods. On the x-axis we show number of weight updates done to the learned optimizer. On the y-axis we show outer-loss. Each point consists of inner-training 100 models, each with five random initializations trained for 10k inner-iterations. We show the average validation performance post-normalization averaged over all tasks, seeds, and inner-training steps. Each line represents a different randomly initialized learned optimizer. In orange we show Adam with a learning rate of $3 * 10^{-5}$ the same value used to train the optimizers in this work. In green, we show Adam with a learning rate of $10^{-3}$, the learning rate that performed best in this 10k outer-iteration regime. In blue we show our learned optimizer.

strategies will perform better. An extensive study of learned optimizer performance on a wide range of state-of-the-art models is a subject for future work.

### 4.6 LEARNED OPTIMIZERS TRAINING THEMSELVES

Finally, we performed an experiment to test if a learned optimizer can be use to train new learned optimizers. Figure 8 shows that this "self-optimized" training curve is similar to the training curve using our hand-tuned training setup (using the Adam optimizer). We interpret this as evidence of unexpectedly effective generalization, as the training of a learned optimizer is unlike anything in the set of training tasks used to train the optimizer. We show outer-training for 10k outer-training iterations, matching the number of inner-iterations used when outer-training the learned optimizer. If outer-training is continued beyond 10k iterations, learned optimizer performance worsens (see Appendix G), suggesting that more inner-iterations are needed when outer-training.

## 5 DISCUSSION

In this work, we train a learned optimizer using a larger and more diverse set of training tasks, better optimization techniques, an improved optimizer architecture, and more compute than previous work. The resulting optimizer outperforms hand-designed optimizers constrained to a single set of hyper-parameters, and performs comparably to hand designed optimizers after a modest hyperparameter search. Further, it demonstrates some ability to generalize to optimization tasks unlike those it was trained on. Most dramatically, it demonstrates an ability to train itself.

The learned optimizer we develop is not without limitations.

**Generalization:** While it performs well for tasks like those in TaskSet, we do not yet fully understand its outer-generalization capabilities. It shows promising ability to generalize to out of distribution problems such as training itself, and ResNet models, but it does not outperform competing algorithms in all settings on simple "out of distribution" optimization tasks, like those seen in Figure 5.

**Optimization / Compute:** Currently it takes significant compute expenditure to train a learned optimizer at this scale, resulting in a nontrivial carbon footprint. Ideally training can be run once and the resulting learned optimizer can be used broadly as is the case with BERT (Devlin et al., 2018).

**Learned optimizer architectures:** We have shown there is considerable improvement to be had by incorporating better inductive biases both in outer-optimization speed and capacity. Improving these architectures, and leveraging additional features will hopefully lead to more performant learned optimizers. Additionally, the current optimizer, while compute efficient, is memory inefficient requiring $> 5\times$ more storage per-parameter than Adam. Modifications similar to those in Shazeer and Stern (2018) will be needed to train larger models.

**Outer-training task distribution:** We have shown that training on larger tasksets leads to better generalization. We have not studied which tasks should be included so as to aid outer-generalization.

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

# A  EXTENDED RELATED WORK

We categorize progress in learned optimizers into three categories: parameterizations, data distributions, and outer-training methodology. We include a review of these previous efforts herein. Finally we include a discussion on other, non-optimizer, learned learning algorithms.

## A.1  PARAMETERIZATIONS

Broadly, authors attack the problem of learned optimization at a high level using different parameterizations.

### A.1.1  CONTROLLER BASED PARAMETERIZATION

One class of learned optimizer parameterizes some function, usually as a neural network, that returns hyperparameters for an existing hand designed method.awk / not precise These parameterizations impose a strong inductive bias towards the hand designed optimizers they parameterize. All inner-learning must be done via manipulation of existing methods. By enforcing a constrained structure, these methods limit the types of learning rules expressible. This limitation does inject strong priors into the model making both outer optimization easier, as well as produce better outer-generalization. These methods often additionally make use of derived features from the currently training model. These features include things like loss values, variance of function evaluations, and variance of weight matrices.

There are a number of existing works that explore different types of features and architectures. Daniel et al. (2016) explores using a simple, linear policy which maps from hand designed features to log learning rate which is then used with SGD, RMSProp or momentum. Xu et al. (2017) learns a small LSTM network with inputs of current training loss, and predicts a learning rate for SGD. Xu et al. (2019) uses an LSTM or an MLP with features computed during training and outputs a scalar which is multiplied by the previous learning rate.

### A.1.2  SYMBOLIC PARAMETERIZATIONS

Existing learning rules are often expressed symbolically by a small number of mathematical expressions operations. All first order optimizers used today are symbolic and expressible by a small collection of mathematical operations. Bello et al. (2017) take inspiration from this and also parameterize optimizers as collections of mathematical primitives. A benifit of this learned optimizer parameterization is that they are more interpretable. For example, Bello et al. (2017) show common subpatterns discovered based on multiplication of the sign of the gradient and sign of the momentum value. Not all computations lend themselves to a symbolic regression. Finding symbolic formula to solve increasingly more difficult tasks starts quickly turns into a combinatorical search and thus difficult to perform (analogous to the difficulties encountered in program synthesis).

### A.1.3  MORE GENERAL PARAMETERIZATIONS

The last family of parameterization is based on directly predicting the update step via more general function approximators. Older work often makes use of simple / linear combinations of features while more recent work makes use of neural networks. There are a few classes of update rule parameterizations.

Bengio et al. (1992) proposes a 7 parameter, and 16 parameter update rule parameterized by mixing various biologically inspired learning signals. Runarsson and Jonsson (2000) use a simple continuous parameterization that operates on error signals and produces changes in inner-parameters.

Recently, there has been renewed interest in learned optimizers, in particular using neural network paramerizations of the learning rules. Andrychowicz et al. (2016) makes use of an LSTM(Hochreiter and Schmidhuber, 1997) possibly with global average pooling to enable concepts like $\ell_2$ gradient clipping to be implemented (Bengio et al., 2013). They also explore Neural Turing Machine (Graves et al., 2014) like parameterizations specifically designed for low rank memory updates so that it can learn algorithms like LBFGS (Nocedal, 1980; Liu and Nocedal, 1989). In contrast, Metz et al. (2019b) makes use of a per-parameter MLP instead of the LSTM.

One critical design choice when designing neural network parameterized learned optimizers is to determine what input features should be used. awk When training neural network models it is critical that the inputs be similar scales for optimization be successful. Andrychowicz et al. (2016) trains on raw gradients which are scaled by either decomposing each scalar into a sign and a magnitude before feeding into an LSTM. Lv et al. (2017) uses the gradient and momentum normalized by the rolling average of gradients squared (similar to the rescaling done by Adam). Wichrowska et al. (2017) also makes use of momentum except using multiple timescales normalized in a similar way. In addition to these momentum terms, Metz et al. (2019b) other features such as weight value and additionally applies a normalization based on the second moment of features.

In Andrychowicz et al. (2016); Lv et al. (2017); Metz et al. (2018) all methods perform updates that are computed independently for each parameters [1]. While expressive, this is expensive as no computation can be shared. Wichrowska et al. (2017) improve upon this by additionally having per-layer, and a global LSTM.

When designing learned optimizer architectures, there are a number of design decisions to keep in mind. One must balance compute cost of the learned optimizer with expressibility. Often this shifts models to be considerably smaller (e.g. <64 hidden units) than most applications of deep learning. Wichrowska et al. (2017), for example makes use of an 8-hidden unit LSTM per-parameter. Selection of features to feed into the learned optimizer is also critical. While traditionally deep learning involves learning every feature, this learning comes at the cost of increased compute which is often not feasible. This paragraph needs some work. First sentence, and "selection of features feed into the learned..."

## A.2    DATA DISTRIBUTION OF TASKS

There are no agreed upon standards when defining datasets for training learned optimizers. Instead the community trains on what ever datasets are available or what ever best approximates a desired goal (training a particular model, creating more general optimizers). Constructing large distributions of tasks is labor intensive and thus not often done.

Metz et al. (2019b) draws inspiration from the few shot learning literature and constructs 10 way classification problems sampling from different classes on imagenet.

Wichrowska et al. (2017) leverages a large distribution of synthetic tasks. These tasks are designed to represent different types of loss surfaces that might be found in loss surfaces.

TaskSet, (Metz et al., 2020) is a dataset of tasks specifically designed for learned optimizer research. We use this dataset throughout our work.

There is a balance between performance of the tasks, and ability to outer-train. Selecting the types of problems we want to train on is often not enough. Additionally, outer-training on the closest task possible will not produce the best optimizer nor even converge. Training on distributions with increased variation smooths the outer-loss surface and makes exploration simpler. This mirrors phenomena found in RL (Heess et al., 2017; Cobbe et al., 2018).

## A.3    OUTER-OPTIMIZATION METHODS

The outer-optimization problem consists of finding a particular set of weights, or configuration of a learned optimizer. A number of different strategies have been proposed.

### A.3.1    HYPER PARAMETER OPTIMIZATION

One of the most common family of outer-learning methods used taken from the family of hyper-parameter optimization methods. In the context of learning optimizers, this can be seen as finding optimizer hyper parameters, e.g. learning rate, over a particular task instance. Numerous methods exist to do this ranging from Bayesian hyper parameter optimization (Shahriari et al., 2015), to grid search, to genetic algorithms. See Golovin et al. (2017) for a more complete description.

---

[1](Metz et al., 2019b) uses normalization across parameters so this is not strictly true

These methods do not work in all settings. The types of outer-learning problems encountered in learned optimizers is different in that the measure of performance (outer-objective) is often an expectation over some distribution of tasks and thus not easily to evaluate. Additionally, the amount of outer-parameters is often larger than simply finding a few hyperparameters. Despite this, we include this here to show the similarity to learned optimizer research.

### A.3.2 REINFORCEMENT LEARNING

Learned optimizers can naturally be cast into a sequential decision process. The state of the system consists of the inner-parameter values, action space is the steps taken, and the reward is achieving a low loss in the future. A number of works have thus taken this viewpoint. Li and Malik (2017a;b) makes use of the guided policy search algorithm. Xu et al. (2019) uses PPO Schulman et al. (2017), Daniel et al. (2016) make use of Relative Entropy Policy Search (Peters et al., 2010). The exact algorithm used is a function of the underlying parameterization.

### A.3.3 NEURAL ARCHITECTURE SEARCH

Instead of learning the policy directly, Bello et al. (2017) makes use of reinforcement learning (PPO (Schulman et al., 2017) to learn a controller which produces the symbolic learned optimizer. This is distinct from §A.3.2 as it does not leverage the sequential nature of the inner problem. Instead, it treats the environment as a bandit problem.

### A.3.4 BACKPROPOGATION / GRADIENT BASED

Computing derivatives through leaning procedures has been explored in the context of hyperparameter optimization in (Bengio, 2000; Baydin and Pearlmutter, 2014; Maclaurin et al., 2015). Andrychowicz et al. (2016) was the first to make use of gradient based learning for this application.

To train a learned optimizer, ideally, one would compute the derivative of final performance with respect to the optimizer parameters. This is often referred to as *unrolling* the entire training procedure into one large graph then running reverse mode automatic differentiation. Not only is this often too expensive to do in practice, the resulting outer-loss surface can be poorly conditioned (Metz et al., 2019b). In practice, approximations are often made.

One common family of approximations is truncated backpropagation through time. The core idea is to break apart the long, sequential unrolled computations into shorter sequences and thus not propagating error back through the entire sequence (Werbos, 1990; Tallec and Ollivier, 2017). This approximation is used widely in neural network parameterized learned optimizers (Andrychowicz et al., 2016; Wichrowska et al., 2017; Lv et al., 2017; Metz et al., 2019b). Unlike in language modeling, truncated backprop has been shown to lead to dramatically worse solutions for meta-learning applications (Wu et al., 2016; Metz et al., 2019b).

A second family of approximations involve not computing any second order or higher terms. This is often refereed to as a"first order" method(Finn et al., 2017; Nichol et al., 2018). Andrychowicz et al. (2016) uses only these first order gradients where as subsequent work, (Wichrowska et al., 2017) computes the full gradient. The trade offs between these two optimization methods has been discussed in the few shot learning literature in the context of MAML / Reptile (Finn et al., 2017; Nichol et al., 2018).

Computing gradients through iterative, non-linear, dynamics has been shown to cause chaotic dynamics. Pearlmutter (1996); Parmas et al. (2018); Maclaurin et al. (2015) showed high sensitivity to learning rate with respect to performance after multiple steps of unrolled optimization. Metz et al. (2019b) shows this issue for neural network parameterized learned optimizers and proposes an alternative objective based on variational optimization(Staines and Barber, 2012) and multiple gradient estimators similar to Parmas et al. (2018).

Despite improvements, gradient based training learned optimizers is still difficult, and requires great care.

### A.3.5 EVOLUTIONARY STRATEGIES

A alternative way to estimate gradients is with black box method such as Evolutionary Strategies(Rastrigin, 1963; Rechenberg, 1973; Nesterov and Spokoiny, 2011; Choromanski et al., 2018; Salimans et al., 2017). These methods are memory efficient, requiring no storage of intermediate states, but can produce high variance estimates. In the case of learned optimizer optimization, however, for certain problems these methods can result in lower variance gradient estimates (Metz et al., 2019b). Hybrid approaches that leverage both gradients and ES have been such as Guided ES (Maheswaranathan et al., 2019) have also been proposed for meta-optimization. This work leverages one of the simplest forms of evolutionary strategies as described in (Salimans et al., 2017) which uses a fixed standard deviation normal distribution for the population.

### A.4 OTHER LEARNED LEARNING ALGORITHMS

There exists a wide variety of other meta-learning / learned learning algorithms. One domain that has received a lot of attention is few shot learning (Vinyals et al., 2016). The methods that perform well usually tend to learn more transferable features, than learning algorithms.

A second domain where learned learning algorithms has been explored extensively is reinforcement learning. We will not survey all methods, but choose three that are most related in the types of challenges they address. Evolved Policy Gradients(Houthooft et al., 2018), explores learning a reinforcement learning algorithm by first learning a loss function then optimizing a policy with respect to this loss. With this parameterization they are able to show generalization to solve novel, and out of distribution tasks. MetaGenRL (Kirsch et al., 2018) is another method focuses on generalizing even further out of distribution enabling transfer across environments with different action and observation spaces. Finally Learned Policy Gradient (Oh et al., 2020) adds learned components to actor critic methods to learn both how the value function is updated, and the policy is updated. They also present transfer results, outer-training on toy environments and transferring to Atari.

## B OUTER OPTIMIZATION DETAILS

In this work, as with Metz et al. (2019b), we use asynchronous, batched training. Each task has a different complexity, thus will produce outer-gradient estimates at a different rate. We use asyncronous minibatched training as synchronous training with these heterogenious workloads would be too slow and wasteful. We tie the outer batch size to the number of workers. To prevent stale gradients, we additionally throw away all outer-gradient estimates that are from more than five outer-iterations away from the current weights.

We optimize all models with Adam and sweep learning rates between 3e-5 and 3e-3 for all experiments. The optimal learning rate changes depending on how long outer-training occurs. We have done a preliminary exploration into learning rate schedules but have not yet been able to improve on this constant schedule. For all outer-training experiments, we always run more than one random optimizer initialization. Due to the relatively small number of units, and biased gradient estimators, performance is dependant on random seed. For all experiments we use gradient clipping of 0.1 applied to each weight independently. Without this clipping no training occurs. This surprises us as our gradient estimator is evolutionary strategies which will not typically have exploding gradients. Upon further investigation, however, the outer-gradient variance is much larger without this clipping.

When computing outer-gradients, we follow Metz et al. (2019b) and compute am outer-loss based on multiple mini batches of data. In our work we use five. Inner-training always uses a single minibatch of inner-training data as well as a single batch of inner-validation data when used.

We first train with 240-360 length unrolls over a max of 10k inner-steps. While training we compute and record inner-training 10k steps for 100 tasks sampled from the outer-training distribution and save outer-parameters every hour. While training we monitor performance across all outer-learning rates and all seeds on the outer-training distribution. When training plateaus, we manually look through these evaluations and select a candidate set of optimizers to further train with an increasing truncation schedule. Not all optimizers fine tune in the same way despite having the same performance on the outer-training data so selecting more than one is critical. At this point we are unsure where this phenomenon comes from.

Next we fine tune theses models with an gradient estimator that does not have short-horizon bias. We explored two methods. The first method is based on an increasing truncation schedule. We tested linearly increasing truncation length from 300-10k inner-steps over the course of 30k or 10k outer-steps. We find the faster increase, 10k steps, performs best. Second, we explored fine tuning with Persisted Evolutionary Strategies – an unbiased gradient estimator (Vicol and et. al., 2020). We found this achieved similar final performance but achieved it in half the time. When fine tuning we also make use of different learning rates. We find higher learning rates make progress faster, but can be unstable in that the performance varies as a function of outer-training step. Additionally, the learning rate chosen depends on the learning rate used previously in the first training phase. Before finetuning, we 'warm up' the Adam internal rolling statistics. While this might not be strictly required, it ensures that there is no decrease in performance early in outer-training. This can be done by simply setting the outer-learning rate to zero for the first 300 outer-iterations.

## C    LEARNED OPTIMIZER ARCHITECTURE DETAILS

In this section we describe the detailed learned optimizer architecture. For ease of understanding we opt to show a mix of pseudo-code based on python and textual descriptions as opposed to mathematical expressions. Finally we chose to describe our optimizer as a series of stateless and pure functions for clarity.

We used this architecture for all of our experiments except of Fig 2b which used an older version of the architecture with additional features which where dropped from the final version.

### C.1    HIGH LEVEL STRUCTURE OF THE OPTIMIZER

The learned optimizer has two main components: a function that maps from some set of inputs, a state, and parameters to some new state and new parameters.

```python
class Optimizer:
    def next_state(inputs: Inputs,
                   state: State,
                   parameters: Params
                   ) -> (State, Params);
```

And a function to produce an initial state from the given inner-parameters.

```python
    def initial_state(self, params: Params) -> State;
```

Parameters are stored as a dictionary of different tensors keyed by name.

```python
Params = Dict[Text, Tensor]
```

For convenience, we also define gradients to be the same type as Params:

```python
Grads = Params = Dict[Text, Tensor]
```

The state consists of multiple values that we will discuss in detail further. For now, however, we list the full state with high level comments.

```python
State = namedtuple(
    "State",
    [
        # current inner training iteration
        "training_step" : Int
         # the statistics for the rolling averages.
        "rolling_features": RollingFeatureState,
        # Hidden state of the lstm
        "lstm_hidden_state": LSTMHiddenState,
         # Activations passed from the MLP to the LSTM.
        "from_mlp": List[Tensor[shape=(from_mlp_size,)]],
        # Activations from LSTM passed to LSTM
```

```
        "from_lstm": Tensor[shape=(from_lstm_size,)],
        # Rolling statistics of the train loss value
        "train_loss_accum": LossAccumState,
         # Rolling statistics of the valid loss value
        "valid_loss_accum": LossAccumState,
         # State to manage inner-gradinet clipping.
        "dynamic_clip": RollingClipState,
         # Parameter values from the nearest 100 steps in the past.
    ])
```

Both from_lstm_size, from_mlp_size are hyper parameters set as part of the learned optimizer to control how much information is sent from the MLP or from the LSTM.

The input to the next_state function consists of inner-gradients, computed on a inner-training batch of data, as well as optionally validation data which is passed in every 10 iterations. We choose to not pass in validation data every steps for computational efficiency.

```
Inputs = namedtuple(
    "Inputs",
    [
        # gradient from task. This is same type as the parameters.
        "inner_grads": Grads,
        # training loss computed from a mini batch.
        "train_loss": float,
        # validation loss from a mini batch of validation data.
        "valid_loss": Optional[float],
    ])
```

Each task specifies a function that samples parameter initialization, as well as to produce outputs.

```
class Task:
    def initial_params(self) -> Params
    def inputs_for_params(self, params) -> (
                Grads, # inner-training gradients
                float, # inner-training loss
                Optional[float]) # optional inner-training validation
```

For outer-training each task also includes a second loss function which computes the task's loss on the outer-validation split of data. Note this uses a *different* validation set of data than the previous function.

```
    def valid_outer_loss(self, params: Params) -> float
```

Inner training / application of the learned optimizer looks like:

```
task = SampleTask()
optimizer = Optimizer()

params = task.initialParams()
state = optimizer.initial_state(state)

for i in range(N): # Length of inner training
    inputs = GetInputFromTask(params, with_valid_data=(i % 10 == 0))
    params, state = optimizer.next_state(inputs, params, state)
```

Computing the outer objective and outer-gradients from inner-initialization looks like the following:

```
task = SampleTask()
optimizer = Optimizer()
# When outer-training with ES, perturb optimizer weights.

params = task.initialParams()
state = optimizer.initial_state(state)
```

```
outer_valid_losses = []
for i in range(N): # Length of a trunction
    inputs = GetInputFromTask(params, with_valid_data=(i % 10 == 0))
    params, state = optimizer.next_state(inputs, params, state)
    for i in range(10):
        outer_valid_losses.append(task.valid_outer_loss(params))

outer_loss = mean(outer_valid_losses)
# When outer-training with ES, one must perturb the optimizer's
                                    parameters before computing
                                    outer_loss.
```

## C.2 UTILITIES / COMPONENTS

First we will describe the individual components and utilities used, then we will go on to describe the full update function.

### C.2.1 ROLLING FEATURES

These are a moving average of gradients and second moments computed similarly to Adam / RMSProp. The rolling state consists of tensors containing momentum values (ms) and second moment values (rms):

```
RollingFeaturesState = collections.namedtuple("RollingFeaturesState",
                                    ["ms": Dict[Text, Tensor],
                                    "rms": Dict[Text, Tensor]])
```

The Text represent names that map to the tensor of the corresponding shape from the parameters. The values are the same shape as the corresponding inner-parameter with an additional axis appended to keep track of multiple different decay values. While its possible to outer-learn these values, we fix them at 0.5, 0.9, 0.99, 0.999, 0.9999.

To update these we construct a helper class.

```
Class RollingState:
    def __init__(self, decay_values):
        self.decay_values = decay_values
```

We define an initial value which is simply simply zeros:

```
    def initial_state(self, params: Params): -> RollingFeatureState
        n_dims = len(decay_values)
        ms = {k: tf.zeros([v.shape + [n_dims]}
        rms = {k: tf.zeros([v.shape + [n_dims]}
        return RollingFeaturesState(ms=ms, rms=rms)
```

To update these values we follow a procedure similar to RMSProp update equations:

```
    def update_state(self, state: RollingFeatureState, grads: Grads) ->
                                    RollingFeatureState:
        ret_ms = {}
        ret_rms = {}
        for k in state.keys():
            g = grads[k]
            s = state[k]

            ret_ms[k] = np.zeros_like(ret.ms[k])
            ret_rms[k] = np.zeros_like(state.rms[k])

            for di, decay in enumerate(self.decay_values):
                ret_ms[k][..., di] = ret[k][..., di] * decay + (1 - decay
                                                ) * grad
```

```
                    ret_rms[k][..., di] = ret[k][..., di] * decay + (1 -
                                                     decay) * grad**2
            return RollingFeatureState(ms=ret_ms, rms=ret_rms)
```

### C.2.2  LOSSACCUM

This represents how we get loss information into our learned optimizer. Loss values have no pre-determined scale and span many orders of magnitude. As such, we must somehow standardize them so that the inputs are bounded and able to be easily used by neural networks. We get around this by keeping track of the normalized mean and variance of the mini-batch losses.

```
LossAccumState = collections.namedtuple("LossAccumState",
              ["mean": float, # rolling mean loss value
               "var": float, # rolling second moment loss value
               "updates": int # Number of updates performed so far
              ])
```

The class that manages these states is parameterized by the decay of the rolling window.

```
class LossAccum:
  def __init__(self, decay):
    self.decay = decay
```

The initial state is simply zeros.

```
  def initial_state(self) -> LossAccumState:
    return LossAccumState(
        mean=tf.constant(0., dtype=tf.float32),
        var=tf.constant(0., dtype=tf.float32),
        updates=tf.constant(0, dtype=tf.int64))
```

To compute updates we do rolling mean and variance computations.

```
  def next_state(self, state: RollingAccumState, loss: float): ->
                              LossAccumState
    new_mean = self.decay * state.mean + (1.0 - self.decay) * loss
    new_var = self.decay * state.var + (
        1.0 - self.decay) * tf.square(new_mean - loss)
    new_updates = state.updates + 1
    return LossAccumState(mean=new_mean, var=new_var, updates=new_updates
                              )
```

Two additional functions are used to normalize loss values for use in neural networks. First, we have a "corrected" mean (similar to what is done by Adam) for a given AccumState.

```
  def corrected_mean(self, state: AccumState) -> float:
    c = 1. / (1 - self.decay**tf.to_float(state.updates) + 1e-8)
    return state.mean * c
```

Second, we have a function that weights one loss by a different AccumState. This is eventually used to weight the validation loss accum against the training loss state allowing the learned optimizer to detect overfitting.

```
  def weight_loss(self, state: AccumState, loss: float) -> float:
    c = 1. / (1 - self.decay**tf.to_float(state.updates) + 1e-8)
    cor_mean = state.mean * c
    cor_var = state.var * c
    l = (loss - cor_mean) * tf.rsqrt(cor_var + 1e-8)
    return tf.clip_by_value(l, -5, 5)
```

### C.2.3 ROLLINGCLIPSTATE

Gradient clipping is a often used technique in deep learning. We found large benefits by applying some form of learned clipping (clipping inner-gradient values) in our learned optimizer. We cannot simply select a default value because gradient norms vary across problem. As such we meta-learn pieces of a dynamic gradient clipping algorithm. This is our first iteration of this concept and we expect large gains can be obtained with a better scheme.

This algorithm is stateful thus also needs some state container.

```
RollingClipState = (int, # the number of times this has been updated,
                     float # rolling average of mean of squared gradient
                                                      values.
                    )
```

This class is parameterized by the decay constant of the rolling average and the multiplier to determine when clipping should start. Both of these values are outer-learned with the rest of the learned optimizer.

```
class RollingGradClip:
  def __init__(self, alpha=0.99, clip_mult=10):
    self.alpha=alpha
    self.clip_mult=clip_mult
```

The initial states are initialized to 1.

```
  def initial_state(self) -> RollingClipState:
    return (tf.constant(1, dtype=tf.float32),
            tf.constant(1.0, dtype=tf.float32)*(1-self.alpha))
```

We provide a normalization function that both updates the RollingClipState and provides clipped gradients.

```
  def next_state_and_normalize(self,
                   state: RollingClipState,
                   grads: Grads) ->
                       RollingClipState, Grads:
    def _normalize(state: RollingClipState, grads: Grads):
        summary_ops = []
        t, snd = state
        clip_amount = (snd / (1-self.alpha**t))*self.clip_mult
        clipgs = [tf.clip_by_value(g, -clip_amount, clip_amount) for g in
                                          grads.values()]
        return clipgs

    t, snd = state
    clipped_grads = self._normalize(state, grads)
    mean_square_list = [tf.reduce_mean(tf.square(g)) for g in
                                  clipped_grads]
    new_snd_moment = tf.sqrt(1e-8 + tf.reduce_mean(mean_square_list))
    next_snd = snd * self.alpha + new_snd_moment * (1. - self.alpha)
    return (t+1, next_snd), clipped_grads
```

### C.3 LEARNED OPTIMIZER SPECIFIC / PUTTING IT ALL TOGETHER

The learned optimizer has two main components. The first is the Optimizer class that manages everything surrounding inner-learning. This function has no learnable outer-parameters. The second is what we call "theta_mod" which contains all of the outer-variables and functions that we are learning.

First we construct the optimizer class with the corresponding theta_mod (see bellow).

```python
class Optimizer:
  def __init__(self,theta_mod):
    self.theta_mod = theta_mod
```

Next the rolling features at fixed, hard coded intervals. These could be outer-learned but in this work they are fixed.

```python
    self.rolling_features = RollingFeatures(
      decays=[0.5, 0.9, 0.99, 0.999, 0.9999])
```

Then the the loss features for use with both training and validation loss. We use a lower decay constant on the validation as it is updated less frequently (once every 10 steps). Once again these could be outer-learned but in this work we leave them fixed.

```python
    self.train_loss_accum = LossAccum(0.95)
    self.valid_loss_accum = LossAccum(0.9)
```

Finally we construct the gradient clipping utility. Unlike the previous parameters, we do outer-learn these and pass in two variables off of the 'theta_mod'.

```python
    self.rolling_clip = RollingGradClip(
                    self.theta_mod.rolling_clip_alpha,
                    self.theta_mod.rolling_clip_mult)
```

Next, we need an initial state for the optimizer defined in C.1. This mostly consists of obtaining initial states from the various components of the learned optimizer.

```python
  def initialState(self, params : Params) -> State:
    shapes = [v.shape.as_list() for v in params.values()]
    return State(
        training_step=0,
        rolling_features=self.rolling_features.initial_state(shapes),
        from_lstm=self.theta_mod.initial_from_lstm(),
        from_mlp=self.theta_mod.initial_from_mlp(len(shapes)),
        lstm_hidden_state=self.theta_mod.initial_rnn_state(len(shapes)),
        train_loss_accum=self.train_loss_accum.initial_state(),
        valid_loss_accum=self.valid_loss_accum.initial_state(),
        dynamic_clip=self.rolling_clip.initial_state(),
    )
```

Next we look to the update performed. We split up into two components – first a validation state, and second a training state.

```python
  def next_params_and_state(self, params: Params,
                            current_state: State,
                            train_loss: float,
                            train_grads: Grads,
                            valid_loss: Optional[float]) -> (Params,
                                                             State):
    if valid_loss:
        current_state = self.next_state_validation(current_state,
                                                   valid_loss)

    params, next_state = self.next_state_trainining(params, current_state
                                        , train_loss, train_grads)
    return params, next_state
```

The validation update simply consists of updating the two components that make use of the loss value.

```python
  def next_state_validation(self, current_state: State,
                            valid_loss: float) -> State:
```

```
next_valid_loss_accum = self.valid_loss_accum.next_state(
    current_state.valid_loss_accum, valid_loss)
return current_state._replace(
            valid_loss_accum=next_valid_loss_accum)
```

The the function applied to the training gradients is much more complicated. Much like the validation features it updates the various rolling statistics including the two loss data structures, the rolling momentum / rms terms, as well as the gradient clipping state.

```
def next_state_trainining(self,
            params: Params,
            current_state: State,
            loss: float,
            grads: Grads) -> (Params, State):
grads = list(grads.values())
next_train_loss_accum = self.train_loss_accum.next_state(
    current_state.train_loss_accum, loss)
next_dynamic_clip, grads = self.rolling_clip.next_state_and_normalize
                                    (current_state.dynamic_clip,
                                     grads)
next_rolling_features = self.rolling_features.next_state(
    current_state.rolling_features, grads)
```

Next, a sequence of features related to loss values are computed.

```
train_loss_feat = self.train_loss_accum.weight_loss(
                                next_train_loss_accum, loss)
valid_loss = self.valid_loss_accum.corrected_mean(current_state.
                                valid_loss_accum)
valid_loss_feat = self.train_loss_accum.weight_loss(
                                next_train_loss_accum,
                                valid_loss)
```

Next we call into the learned optimizer function in "theta_mod" which does the bulk of the computation.

```
next_params, from_lstm, from_mlp, lstm_hidden_state = \
  self.theta_mod.compute_update_and_next_state(
        train_loss_feat, valid_loss_feat,
        current_state.from_mlp,
        current_state.from_lstm,
        current_state.lstm_hidden_state,
        next_rolling_features,
        grads,
        params,
        current_state.training_step,
        )
```

We then populate the next state with all the outputs.

```
 next_state = State(
    rolling_features=next_rolling_features,
    training_step=current_state.training_step + 1,
    lstm_hidden_state=lstm_hidden_state,
    from_lstm=from_lstm,
    from_mlp=from_mlp,
    train_loss_accum=next_train_loss_accum,
    valid_loss_accum=current_state.valid_loss_accum,
    dynamic_clip=next_dynamic_clip,
 )

 return next_params, next_state
```

### C.4 "THETAMOD": THE LEARNED OPTIMIZER OUTER-PARAMETERS

All of the meta-learned outer-parameters are stored in a class called ThetaMod. Its constructor defines various dimensions of the different Sonnet modules.

```python
class ThetaMod:

  def __init__(self,
               # Size of the features from the per parameter mlp.
               from_mlp_size=16,
               # size of the features from the per tensor LSTM.
               from_lstm_size=18,
               # Size of features from the lstm being passed to the per
                                               param MLP.
               lstm_to_ff=17,
               # LSTM hidden size.
               lstm_hidden_size=64,
               # Multiplier on output to improve conditioning
               step_multiplier=0.001,
               # Multiplier on output to improve conditioning
               magnitude_rate=0.001,
               **kwargs):
    self.step_multiplier = step_multiplier
    self.magnitude_rate = magnitude_rate
```

This constructor creates a shared sonnet module for the per layer LSTM. This is constructed here as it is needed in more than one method.

```python
    self.rnn = InitialStateLSTM(lstm_hidden_size)
```

Second, we construct the per parameter feed forward network. This is a tiny network that operates on each parameter. It outputs two values for producing a step, then from_mlp_size more values which are fed back into the LSTM.

```python
    self.ffmod = snt.nets.MLP([32, 32] + [2 + self.from_mlp_size], name
                              ="PerParamMLP")
```

Third, we construct two linear projections that map from the RNN output to either the per parameter feed forward network, or back to the global features.

```python
    self.rnn_to_ff = snt.Linear(lstm_to_ff, name="rnn_to_ff")
    self.rnn_to_in = snt.Linear(from_lstm_size, name="rnn_to_in")
```

Finally, we initialize the two variables used by the rolling gradient clipping terms. We parameterize these variables on a log scale (possibly subtracting 1) so as to roughly match the scaling of the other neural network weights.

```python
    def get_log_param_variable(name,
                               initial_value,
                               minus_one=False):
      if minus_one:
        initial_value = 1 - initial_value
      init_log = np.log(initial_value)
      v = tf.get_variable(
          name,
          initializer=tf.ones(shape=[]) * tf.constant(
              init_log, dtype=tf.float32),
              trainable=True)
      ret = tf.exp(v)
      if minus_one:
        ret = 1 - ret
      return ret
```

```python
        self.rolling_clip_alpha = get_log_param_variable(
                name="rolling_clip_alpha",
                initial_value=0.99,
                minus_one=True)

        self.rolling_clip_mult = get_log_param_variable(
                name="rolling_clip_mult",
                initial_value=10.0,
                minus_one=False)
```

The heart of this function takes the features passed in from the optimizer, applies some neural network based learning algorithm, then produces the next weight value as well as various pieces of hidden information for the next iteration.

```python
    def compute_update_and_next_state(self,
        # Training loss feature
        train_loss_feat: Tensor[shape=()],
        # Validation loss feature
        valid_loss_feat: Tensor[shape=()],
        # Information passed to each layer of the LSTM from the per param
        #                                  MLP.
        from_mlp: List[Tensor[shape=(from_mlp_size,)]],
        # Information aggregated across LSTM layers to be passed to all
        #                                  LSTM layers.
        from_lstm: Tensor[shape=(from_lstm_size,)],
        # The current LSTM state.
        lstm_state: LSTMHiddenState,
        # Features from the momentum / rolling second moments.
        rolling_features: RollingFeatureState,
        # Gradients from current training mini-batch
        grads: Grads,
        # Current parameters
        params: Params,
        # current inner-iteration
        training_step: int,
        ) ->
            # the output parameter value.
            (Params,
            # next from_lstm
            Tensor[shape=(from_lstm_size,)],
            # next from_mlp
            List[Tensor[shape=(from_mlp_size,)]],
            # next lstm hidden state.
            LSTMHiddenState):
```

We first compute the global feature vector. These consist of features related to the loss curves, as well as the aggregated data from the previous LSTM execution. These features form a rank 1 tensor.

```python
    global_features = self.compute_global_features(train_loss_feat,
                                                   valid_loss_feat,
                                                   from_lstm)
```

Next we compute the per tensor features. This is a rank two tensor, consisting of the number of tensors, and a feature dimension.

```python
    per_tensor_features  = self.compute_tensor_features(grads,
                                                        params,
                                                        from_mlp,
                                                        rolling_features)
```

We tile the global features to be num tensors by num global features then concat these to the per tensor features and pass them into the per layer LSTM.

```
rnn_inputs = tf.concatenate([per_tensor_features,
        tf.tile(global_features, [per_tensor_features.shape[0], 1])])
lstm_out, next_lstm_hidden_state = self.rnn(rnn_inputs, lstm_state)
```

We then linearly project the LSTM outputs to be input to the per parameter feed foward network, and unstack them to form a a list of rank 1 features – one per tensor.

```
ff_inputs = self.rnn_to_ff(lstm_out)
ff_inputs = tf.unstack(ff_inputs)
```

Next we iterate over each parameter, and first compute per parameter features. The per parameter features consist of gradients, parameter values, parameter values 100 steps in the past, momentum values, second moment values, activations from the per tensor LSTM. Additionally there are two global features included: the training step and the number of tensors total. Two empty lists are initialized to accumulate results.

```
next_to_lstms = []
next_params = []

for wi in range(len(params)):
    feats = compute_per_param_features(grads[wi],
                                       param[wi],
                                       rolling_features.ms[wi],
                                       rolling_features.rms[wi],
                                       ff_inputs[wi]
                                       training_step,
                                       len(params))
```

We pass these per parameter features through these features into a per parameter MLP which produces a single output.

```
output = self.ffmod(feats)
```

We split this output into three pieces: A direction, a magnitude, and features which are be fed back to the per tensor LSTM after being reduced along the parameter dimension.

```
direction = output[:, 0:1]
magnitude = output[:, 1:2]
to_lstm = tf.reduce_mean(output[:, 2:], axis=0)
```

A step can then be computed by exponentiating the magnitude and multiplying by a direction. Rescaling these quantities is critical. Without this rescaling, the magnitude of the updates is extremely large, resulting in chaotic training dynamics which makes outer-learning difficult.

```
step = direction * tf.exp(magnitude * self.magnitude_rate) * self
                                        .step_multiplier
```

We then subtract step from the previous parameter value. While it is possible just to predict a new parameter directly, learning deltas is significantly easier to learn and results in much more stable training.

```
next_param = param[wi] - step
```

Finally, we collect all the information needed to be returned from the per parameter LSTM and output.

```
next_params.append(next_param)
next_to_lstms.append(next_to_lstm)
```

With the for loop over, we then return the necessary info. Before this, however, we must aggregate information from the per tensor LSTM to be able to pass it to the next layer's global features after applying a linear projection from the LSTM output and reducing along the tensor dimension.

```
    from_lstm = tf.reduce_mean(self.rnn_to_in(lstm_out), axis=0)
    return next_params, from_lstm, next_to_lstms, next_lstm_hidden_state
```

Now we will discuss the different features methods. First, computing the global features.

```
def compute_global_features(self,
                            train_loss_feat: Tensor[shape=()],
                            valid_loss_feat: Tensor[shape=()],
                            from_lstm: Tensor[shape=(n_lstm_out,)]
                            ): -> Tensor[shape=(n_global_features,)]
```

This function simply returns the concatenation of the Two loss features, and the aggregated features from the per layer lstm.

```
    return tf.concatenate([train_loss_feat, valid_loss_feat], from_lstm)
```

Next we look to the per tensor features. We return a 2D tensor with the leading dimension a "batch size" being the number of tensors, and the second dimension representing the feature dimension.

```
def compute_tensor_features(self,
        grads: Grads,
        params: Params,
        from_mlp: List[Tensor[shape=(from_mlp_size,)]],
        rolling_features: RollingFeatureState
        ): -> Tensor[shape=(n_tensors, n_features)]
```

We compute a large number of features. At this point have not extensively ablated each one and we expect many are duplicate or unneeded. Before we begin listing features, however, we make use of a helper function that computes the clipped log abs of tensors.

```
    def clip_log_abs(v):
      mag = tf.log(1e-8 + tf.abs(v))
      return tf.clip_by_value(mag, -5, 5)
```

We begin by iterating over each tensor.

```
stacked_inputs = []
for i in range(len(params)):
  inputs = {}
```

First, we compute features based on the rolling momentum values. We compute features that represent the log magnitude, the sign, and the log variance.

```
    mean_ms = tf.reduce_mean(rolling_features.ms[i])
    inputs["mean_ms_mag"] = clip_log_abs(mean_ms)
    inputs["mean_ms_sign"] = tf.sign(mean_ms)
    var_ms = tf.reduce_mean(tf.square(rolling_features.ms[i] - mean_ms)
                                        )
    inputs["var_ms"] = clip_log_abs(var_ms)
```

We compute a similar set of features for the second moment features.

```
    mean_rms = tf.reduce_mean(rolling_features.rms[i])
    inputs["mean_rms"] = clip_log_abs(mean_rms)
    inputs["mean_sign"] = tf.sign(mean_rms)
    var_rms = tf.reduce_mean(tf.square(rolling_features.rms[i] -
                                        mean_rms))
    inputs["var_rms"] = clip_log_abs(var_rms)
```

A similar set for the inner-parameters:

```
    v = params[i]
    mean_v = tf.reduce_mean(v)
    inputs["mean_v_mag"] = clip_log_abs(mean_v)
    inputs["mean_v_sign"] = tf.sign(mean_v)
    var_v = tf.reduce_mean(tf.square(v - mean_v))
    inputs["var_v"] = clip_log_abs(var_v)
```

Next we include the magnitude of the weight norm.

```
    inputs["norm_weight"] = clip_log_abs(tf.norm(params[i]))
```

As well as the gradient norm:

```
    inputs["g_norm"] = clip_log_abs(tf.norm(grads[i]))
```

Next we add a feature if the tensor is a scalar or not.

```
    if len(v.shape.as_list()) == 0:
      is_scalar = tf.constant(1.0)
    else:
      is_scalar = tf.constant(-1.0)
    inputs["is_scalar"] = is_scalar
```

The shape of the underlying tensor is also a useful feature. The length of this shape varies though depending on the rank. As such we first pad the tensor to be at least rank 4 and then pass the log shape shifted by -1. This shift is to keep this value roughly zero mean.

```
    extra_dims = [1.] * (4 - len(v.shape.as_list()))
    shape_stack = tf.concat([tf.to_float(tf.shape(v)),
                             tf.stack(extra_dims)],
                            axis=0)

    for j in range(4):
      # shift so 1D is neg one to be better scaled.
      inputs["shape_%d" % j] = tf.log(shape_stack)[j] - 1.0
```

The features from the previous execution's MLP are included. These are to enable communication back from the MLP to the per layer network.

```
    inputs["from_mlp"] = from_mlp[i]
```

This dictionary of features is then flattened (adding appropriate dimensions if needed) and appended to the list of inputs.

```
    values = sorted_values(inputs)
    reshaped = [
        tf.expand_dims(v, 0) if len(v.shape.as_list()) == 0 else v
        for v in inputs.values()
    ]
    stacked_inputs.append(reshaped)
```

Once the stacked_inputs is filled, we zip and stack and concat the values forming a number of layers by number of features tensor.

```
  stacked_features = [tf.stack(v) for v in zip(*stacked_inputs)]
  return tf.concat(stacked_features, axis=1)
```

The last set of features is the per parameter features.

```
  def compute_per_param_features(self,
            grad: Tensor,
```

```
                    param: Tensor,
                    ms: Tensor,
                    rms: Tensor,
                    ff_inputs: Tensor[shape=(num_ff_inputs,)],
                    training_step: int,
                    num_params: int
                    ): -> Tensor[shape=(num_params, num_features)]
```

Many of these inputs are of the shape of the underlying tensor. Because many of these operations are per parameter we construct a flat version of all of these.

```
        flat_g = tf.reshape(grad, [-1, 1])
        flat_v = tf.reshape(param, [-1, 1])
        flat_ms = tf.reshape(ms, [-1, ms.shape[-1]])
        flat_rms = tf.reshape(rms, [-1, rms.shape[-1]])
```

As with the per layer features, we add a series of features. First we add raw features for the gradients, parameter values, momentum, and second moment accumulators, and past weights.

```
        inps = {}
        inps["flat_g"] = flat_g
        inps["flat_v"] = flat_v
        inps["m"] = m
        inps["rms"] = rms
```

We then add some derived features such as the log abs parameter value:

```
        inps["log_abs_v"] = tf.log(tf.abs(flat_v) + 1e-8)
```

Finally we add terms based on the rsqrt of the second moment terms inspired by how Adam normalizes.

```
        rsqrt = tf.rsqrt(rms + 1e-6)
        rms_scaled_g = m * rsqrt
        inps["rms_scaled_g"] = rms_scaled_g
        inps["rsqrt"] = rsqrt
```

These values are concatenated and normalized by a second moment normalizer computed across the num tensor dimension. We additionally clip the output of this normalizer. This is crucial for stable training as some of these features will produce large magnitudes in some channels.

```
        inp = tf.concat(sorted_values(inps), 1)
        inp = inp * tf.rsqrt(1e-8 +
                        tf.reduce_mean(tf.square(inp), axis=0,
                                                        keep_dims=
                                                        True))
        inp = tf.clip_by_value(inp * 0.5, -1, 1)
```

Next, we embed the current inner-training step with sinusoids of different frequencies.

```
        def sin_embedding(x):
          mix_proj = []
          for i in [1, 3, 10, 30, 100, 300, 1000, 3000, 10000, 30000,
                                        100000]:
            s = tf.to_float(tf.to_float(i) / float(np.pi))
            mix_proj.append(tf.sin(s * tf.to_float(x)))
          return tf.stack(mix_proj)

        step = utils.sin_embedding(training_step)
        stack_step = tf.tile(
            tf.reshape(step, [1, -1]),
            tf.stack([tf.shape(flat_g)[0], 1]))
```

We compute a features based on the number of tensors:

```
log_num_tensors = tf.log(float(len(grads_and_vars))) - 1.

stack_num_tensors = tf.tile(
    tf.reshape(log_num_tensors, [1, 1]),
    tf.stack([tf.shape(flat_g)[0], 1]))
```

As well as features about the number of parameters in the current tensor:

```
log_n_weight = tf.log(tf.to_float(tf.shape(flat_v)[0]))
stack_log_n_weight = tf.tile(
    tf.reshape(log_n_weight, [1, 1]),
    tf.stack([tf.shape(flat_g)[0], 1]))
```

Next, we compute some statistics about the log norm of the weight matrix:

```
log_norm = tf.log(tf.norm(flat_v) + 1e-8)
stack_log_norm = tf.tile(
    tf.reshape(log_norm, [1, 1]),
    tf.stack([tf.shape(flat_g)[0], 1]))
```

Next, we tile the input from the LSTM so as to add a number of parameters dimension.

```
ff_inp = tf.tile(
    tf.reshape(ff_inputs[i], [1, -1]),
    tf.stack([tf.shape(flat_g)[0], 1]))
```

Finally we concat and return all these features.

```
return tf.concat([
    inp, stack_step, stack_num_tensors, stack_log_norm,
    stack_log_n_weight, ff_inp
], axis=1)
```

We have discussed how new values of LSTM hidden state, activations from the LSTMs, and activations from the MLP are produced, but we have not yet shown the initial values. In all cases they are initialized from outer-parameterized values shown bellow.

```
@snt.reuse_variables
def initial_rnn_state(self, n):
  s = self.rnn.initial_state(n, trainable=True)
  return s

@snt.reuse_variables
def initial_from_lstm(self):
  return tf.get_variable(
      name="initial_from_lstm",
      shape=[self.from_lstm_size],
      dtype=tf.float32,
      trainable=True)

@snt.reuse_variables
def initial_from_mlp(self, n):
  s = tf.get_variable(
      name="initial_from_mlp",
      shape=[self.from_mlp_size],
      dtype=tf.float32,
      trainable=True)
  return [s for _ in range(n)]
```

# D    ADAM8P

This matches the adam8p optimizer described in (Metz et al., 2020). The eight hyper-parameters are: the learning rate, $\alpha$, first and second moment momentum, $\beta_1$, $\beta_2$, the numerical stability term, $\epsilon$, $\ell_2$ and $\ell_1$ regularization strength, and learning rate schedule constants $\lambda_{\text{exp\_decay}}$ and $\lambda_{\text{linear\_decay}}$.

$$\phi^{(0)} = \text{problem specified random initialization} \tag{1}$$

$$m^{(0)} = 0 \tag{2}$$

$$v^{(0)} = 0 \tag{3}$$

$$g^{(t)} = \frac{d}{d\phi^{(t)}}(f(x;\phi^{(t)}) + \ell_2||\phi^{(t)}||_2^2 + \ell_1||\phi^{(t)}||_1) \tag{4}$$

$$m^{(t)} = \beta_1 m^{(t-1)} + g^{(t)}(1 - \beta_1) \tag{5}$$

$$v^{(t)} = \beta_2 v^{(t-1)} + (g^{(t)})^2(1 - \beta_2) \tag{6}$$

$$\hat{m}^{(t)} = \frac{m^{(t)}}{1 - \beta_1^{t+1}} \tag{7}$$

$$\hat{v}^{(t)} = \frac{v^{(t)}}{1 - \beta_2^{t+1}} \tag{8}$$

$$u^{(t)} = \frac{\hat{m}^{(t)}}{\sqrt{\hat{v}^{(t)}} + \epsilon} \tag{9}$$

$$s_{\text{linear}}^{(t)} = \max(1 - t\lambda_{\text{linear\_decay}}, 0) \tag{10}$$

$$s_{\text{exp}}^{(t)} = \exp(-t\lambda_{\text{exp\_decay}}) \tag{11}$$

$$\phi^{(t+1)} = \alpha s_{\text{linear}}^{(t)} s_{\text{exp}}^{(t)} u^{(t)} \tag{12}$$

We sample learning rate logritmically between 1e-8 and 10, beta1 and beta2 we parametrize as $1 - x$ and sample logrithmically between 1e-4 and 1 and 1e-6 and 1 respectively. For learning rate schedules we sample linear decay between 1e-7, 1e-4 logrithmically and exponential decay logrithmically between 1e-3, 1e-6. We sample both $\ell_1$ and $\ell_2$ logrithmcally between 1e-8, 1e1.

# E    PERFORMANCE TABLE

We show numerical performance measurements of our learned optimizer and baseline optimizers. Each value is the mean or median over 100 different inner-training tasks.

| Optimizer | train | | test | |
|---|---|---|---|---|
| | mean | median | mean | median |
| learned | 0.077325 | 0.024167 | 0.113242 | 0.028928 |
| global adam lr (1 Trial) | 0.226807 | 0.105513 | 0.261613 | 0.092532 |
| global adam8p, RS (1 Trial) | 0.240924 | 0.109519 | 0.257368 | 0.100959 |
| global nadamw, RS (1 Trial) | 0.215176 | 0.098448 | 0.229359 | 0.086703 |
| per task adam lr (14 Trial) | 0.150443 | 0.067373 | 0.152028 | 0.063194 |
| per task opt_list (10 Trial) | 0.082756 | 0.039647 | 0.086195 | 0.038541 |
| per task opt_list (100 Trial) | 0.041560 | 0.028851 | 0.047224 | 0.031042 |
| per task opt_list (1k Trial) | 0.039348 | 0.028254 | 0.043193 | 0.029897 |
| per task adam8p (10 Trial) | 0.278858 | 0.132965 | 0.264282 | 0.128594 |
| per task adam8p (100 Trial) | 0.095372 | 0.046926 | 0.107177 | 0.046672 |
| per task adam8p (1k Trial) | 0.044144 | 0.028692 | 0.060663 | 0.028813 |
| per task adam8p + opt_list(2k Trial) | 0.018403 | 0.023350 | 0.025419 | 0.025555 |

## F    EXPERIMENTAL DETAILS FOR FIGURES

### F.1    OUTER TRAINING OF DIFFERENT LEARNED ARCHITECTURES

All models are trained with truncated evolutionary strategies using the parameters described in the main text. Instead of using the two stage schedule we only train with truncation's of size 240-360. We use the same, fixed learning rate for the FF and LSTM_FF model of 0.000500. For LSTM we found a lower learning rate of 0.000100 to perform better. Ideally we would include an extensive hyperparameter comparison, but the computational cost is prohibitive.

For the FF model we make use of a two hidden layer 32 unit MLP. In addition to the features described in (Metz et al., 2019b) we use RMS terms which improve performance over that reported in (Metz et al., 2019b) by a small amount.

For the L2LBGDGD model we copy Andrychowicz et al. (2016) using a two hidden layer, 20 unit GRU (Chung et al., 2014) with the same input gradient processing.

For the LSTM model, we make use of a model inspired by (Andrychowicz et al., 2016). We make use of a 64 dimension LSTM hidden size to produce two outputs, a direction and a sign which are combined similar to (Metz et al., 2019b).

Performance on 100 tasks (each with five random seeds) is recorded over the course of training. We show an exponential moving average of this data in the figure.

### F.2    OUTER TRAINING WITH DIFFERENT SIZED MODELS

All models are trained with truncated evolutionary strategies using parameters described in the main text. Instead of using the two stage schedule we only train with truncation's of size 240-360. We use a learning rate of $3 * 10^{-3}$ for all models.

### F.3    LARGE SCALE TRANSFER WITH RESNET MODELS

We make use of two ResNetV2 models. Before applying the residual blocks, we pass the inputs through a 64 unit, 7x7 convolutional kernel with stride 2 followed by max pooling of size 3 with a stride of 2. This aggressive down sampling was originally designed for larger image models but was also left in despite our small sized models.

We then apply some number of residual blocks parameterized by a (number of output channels, number of bottleneck channels, stride).

At the output we apply batch norm, reduce over the spatial dimensions, and project to the the appropriate number of classes.

Our goal with these experiments are not to seek peak performance. Instead show that our learned optimizer is capable of generalizing to vastly different distributions of models.

**CIFAR-10 Resnet**    This model makes use of four residual blocks: [(128, 32, 1), (128, 32, 2), (128, 64, 1), (128, 64, 2)].

**Imagenet Resnet**    This model makes use of 11 residual blocks: [(128, 32, 1), (128, 32, 2), (256, 64, 1)*2, (256, 64, 2), (512, 128, 1)*3, (512, 128, 2), (1024, 256, 1)*2].

### F.4    SELF OPTIMIZATION DETAILS

For our baseline learned optimizer trained with Adam, we search over two learning rates, $10^{-5}$ and $3 * 10^{-3}$ and select the best one (1e-3) for the given 10k outer-training steps as well as selected the optimizer used to train the best optimizer (3e-5).

When training an optimizer with a new optimizer there are no hyper parameters. We found we must also include the original gradient clipping used for the Adam models (clipping all values between -0.1, 0.1). Without this, our learned optimizer is not able to make any progress. Understanding this is

an interesting to us as we would have suspected the dynamic inner-clipping should have addressed this.

For technical reasons, the learned optimizer does not have easy access validation losses. Instead, we use the training loss for both. This is computed as the mean over the normalized losses computed over a single batch.

## G  SELF OPTIMIZATION EXTENDED

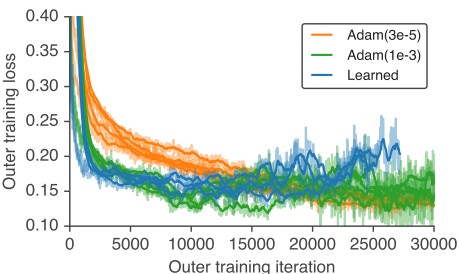

Figure 8: When using a learned optimizer to train itself, we find good performance early in training, but diverge after 10k outer-iterations. On the x-axis we show number of weight updates done to the learned optimizer. On the y-axis we show outer-loss. Each point consists of inner-training 100 models, each with five random initializations trained for 10k inner-iterations. We show the average validation performance post-normalization averaged over all tasks, seeds, and inner-training steps. Each line represents a different randomly initialized learned optimizer. In orange we show Adam with a learning rate of 3e-5 the same value used to train the optimizers in this work. In green, we show Adam with a learning rate of 1e-3, the learning rate that performed best in this 10k outer-iteration regime. In blue we show our learned optimizer. Light colors represent instantaneous outer-losses while solid denotes smoothing done via an exponential moving average.

## H    INFRASTRUCTURE USED

The types of learning systems we discuss here are more complex than the average deep learning training setup. We are training not one, but many different kinds of models on different datasets with wildly different properties. As a result, the static graph paradigm of traditional deep learning models – distributed one graph and using MPI / allreduce style primitives – is insufficient. Finally, evaluating the performance of an optimizer is not a trivial task as it requires inner-training a large number of different models for a large number of inner-iterations.

Developing custom training and evaluation infrastructure has been critical to the success of this family of work. We describe the training infrastructure employed, discuss techniques we use for evaluation and monitoring. At this time, we are unable to release a fully running open source implementation due to the use of internal tools.

### H.1    OUTER-TRAINING CLUSTER

The cluster consists of six kinds of jobs – workers, a learner, gradient storage, summary aggregators, evaluation chief, evaluation worker. Additionally we make use of a distributed file system (Ghemawat et al., 2003).

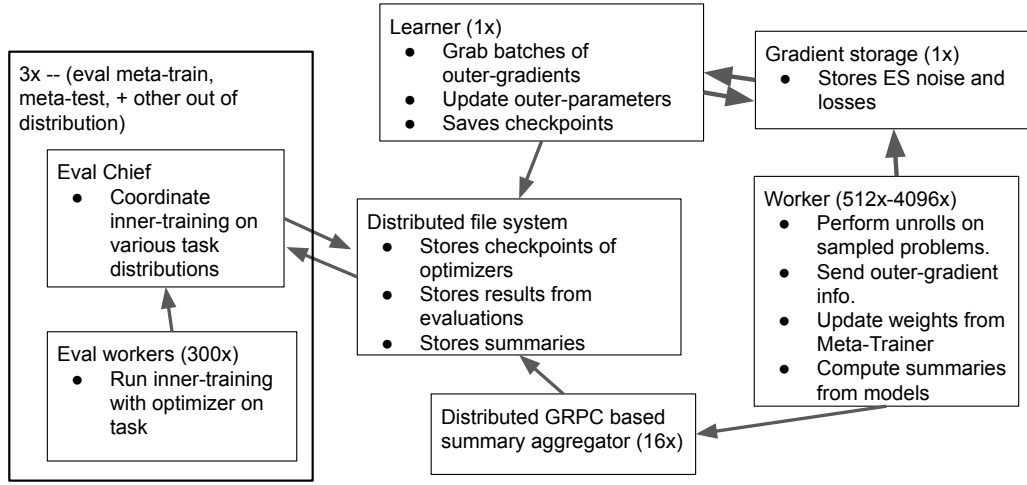

Figure 9: High level / system diagram for infrastructure used. See Appendix H for more information.

**Workers**    Workers are jobs that sample a outer-task, build a tensorflow (Abadi et al., 2016) graph and provides some way to estimate outer-gradients. In this work, we leverage antithetic ES sampling with shared randomness where possible. In practice, this means we start two inner-training procedures starting from the same initializations, using the same minibatches of data, but leveraging two slightly differnt outer-parameter values. Before each outer-gradient estimate new outer-parameters, as well as the current outer-iteration, are obtained by RPC call to the learner. This gradient estimate, the current outer-iteration (e.g. global step) along with the type of problem selected is sent to the key value storage worker keyed with some unique identifier.

**Gradient Storage**    The gradient storage job is a distributed dictionary which stores information computed from the worker. At this point, given the low frequency of updates only a single machine is required. This is a separate machine from the learner to control the number of queries required sent to the learner.

**Learner**    The learner polls the gradient storage for batches of data. If the outer-training iteration is within some fixed amount of time to the current outer iteration the data is deleted. In this work we use 5. This prevents stale gradients from being used to update the outer parameters. This has the negative effect of potentially ignoring the slowest tasks. At this point, however, we find that this rejection threshold is quite small (<1%) and thus safely ignored. When a batch of outer-information is obtained, the learner aggregates outer-gradients, and performs an update of outer-optimization, in our case Adam. Current loss values for each task family are also recorded in a dictionary, and aggregated across all tasks seen thus far. We find this averaged signal, as well as simply just logging out the loss from the current batch, to be useful for evaluating model performance as it is lower variance and is stationary in time. This job is also responsible for writing the meta-parameters to disk every 10 min for use later and for evaluation.

**Summary aggregators**    When training a standard model, users monitor values – e.g. activation values, losses, or accuracies. In this work we are training thousands of different neural network tasks at a time and would ideally like to be able to obtain more insight into what is going on. Existing tooling such as TensorFlow's event files are inadequate.

We work around this by introducing a distributed service meant just for summary aggregation. This service exposes a RPC api that takes batches of summaries and logs them to disk in a custom binary format for fast visualization later. We shard request based on the name of the summary. We additionally log out subsampled data for ease of viewing after training.

In this work, we process roughly 10k summaries a second split across 24 machines. We found being able to do in-depth post-hoc analysis of model training to be critical to diagnosing issues and uncovering bugs. For example, looking at gradient variance for each task in the outer-training set let us diagnose which tasks are diverging and why. To ensure that summary computation does not cause increased computational overhead we make use of both stochastic logging as well as both batching of RPC calls to summary aggregators, and sending these RPC in parallel with computation.

**Evaluations:**    Evaluating a learned optimizer is not nearly as simple as say performing forward passes on some test set of minibatches. Evaluation consists of an expectation over training many different tasks. To obtain a low variance estimate we employ a large amount of compute to evaluate the checkpoints saved out by the learner. To lower variance, we pick a fixed set of tasks to evaluate. In addition to the meta-training task distribution (which we pick a subset of 100 to evaluate), we also evaluate on a test set of tasks which is iid to the meta-train but never seen when outer-training as well as an extremely out of distribution set of data not close to tasks in the outer-training distribution. By monitoring a variety of different problems, with different similarities to the meta-training set we obtain a better sense of outer-generalization. These evaluations take time, around an hour of compute per task, so it must parallelized.

Originally, we tried to avoid this complexity and obtain some measure of outer-loss directly from the training cluster. While this works to some extent, it is noisy, and doesn't capture what we actually care about – performance of some fixed weights of a learned optimize when applied to a target problem for an extended period of time. Averaging over the meta-training cluster also doesn't capture variance from outer-weight to outer-weight.

The evaluation of one task also requires orchestration. We employ a evaluation chief, and evaluation workers for each task set. While its possible to do this type of evaluation offline, we find running online yields a much better workflow and thus speeds up research.

**Evaluation Chief:**    The evaluation chief is responsible for watching directories on a distributed filesystem and to enqueue evaluation tasks. Workers use RPC to request, and to send back results (e.g. learning curves). When a set of tasks is complete for a given outer-parameter checkpoint the results are written out to disk for later analysis.

**Evaluation workers:**    Evaluation workers request task configurations from the chief over RPC. This configuration is parsed and converted to a TensorFlow graph which is then run. Over the course of running, various signals including training loss, validation loss, and test loss, are recorded. Upon completion these results are sent back over RPC to the chief to be aggregated and written to disk.

## H.2 COMPUTATIONAL EXPENSE

At this point, training of a learned optimizer is quite expensive. It roughly entails 60K CPU cores for around a month, or around 5k CPU years. Much as neural architecture search has been dramatically optimized and improved, we hope learned optimizers will obtain a similar outer-training speedups. The energy and environmental impact of this work is also worth noting. These models take on the order of 200 megawatt hours of power to outer-train. We believe that these methods can be used to speed up development of many existing models once outer-trained. By meta-training a hyper parameter free model, one has to do no hyper parameter tuning of the optimizer for example. In general, one should weight the costs of outer-training against the potential savings achieved. In the future we expect these models will be trained once and applied everywhere.

Finally, we would like to highlight that all of this work was performed on CPU. In tests on GPU (without optimizing) we found performance to be similar. Modern accelerator hardware, TPU, GPU, are ill-suited for the small workloads we perform here (serially training lots of small networks). We believe it is possible to vectorize the inner-training of neural networks (e.g. training N networks instead of just 1 per worker) but have not explored this route yet.

