# OpenReview forum: "Overcoming barriers to the training of effective learned optimizers"
_ICLR.cc/2021/Conference — Reject_

### Official Review · AnonReviewer4 · 2020-10-29
**Official Blind Review #4**

**Rating:** 7
**Confidence:** 3

**Review:**

The goal of a learned optimizer is to replace a human-designed optimizer with a parametric optimizer. However, prior learned optimizers were ineffective at generalizing to a diverse set of tasks. This paper investigates how to learn a useful optimizer by increasing computational scale, building a large, diverse training dataset, and designing the learned optimizer's architecture.

Strengths:
+ This paper thoroughly examines the challenges of how to train a learned optimizer.

Weaknesses:
+ Training the learned optimizer is fairly complex and computationally expensive, which will prevent broader adoption and makes the paper difficult to reproduce.

Questions:
1) What conditions are necessary for a learned optimizer to replace a standard optimizer like Adam?
2) Could a learned optimizer work with noisy, quantized gradients? e.g. deploying a learned optimizer in a federated learning environment

---

> ### Author Response · Authors · 2020-11-19
> **Response**
>
> Thank you for your time and thoughtful review.
>
> \> "complex and computationally expensive, which will prevent broader adoption and makes the paper difficult to reproduce."
>
> This is a good observation. Much in the same way that neural architecture search originally required extreme compute resources, but became more computationally efficient with one-shot methods, we are exploring ways to make this line of research more accessible. (While not relevant to this paper, we have preliminary results achieving two+ orders of magnitude speedup using accelerators and ML compiler tech -- it is possible to take advantage of similarities in the learned optimizer updates on different tasks, and compute parameter updates for multiple tasks using monolithic multi-task tensor-update operations.)
>
> We also want to emphasize that while learned optimizers take large amounts of compute to train, they are roughly as expensive as hand designed optimizers to apply. If one team trains a high performance learned optimizer, that optimizer can be used by the entire field without onerous compute.
>
> \> "What conditions are necessary for a learned optimizer to replace a standard optimizer like Adam?":
>
> This is somewhat subjective, but we believe a minimum requirement is that our optimizers reliably achieve a better loss in less wallclock time than can be achieved by a hand designed method with a moderate amount of hyperparameter tuning. We have not yet achieved this goal, but we believe current work is getting close.
>
> \> "Could a learned optimizer work with noisy, quantized gradients? e.g. deploying a learned optimizer in a federated learning environment"
>
> Yes on all counts. We expect you would need to train specifically to perform well in these domains though.

---

### Official Review · AnonReviewer2 · 2020-10-29
**Interesting large-scale analysis of learned optimizers, but unfortunately no barriers are overcome**

**Rating:** 4
**Confidence:** 3

**Review:**

Summary

This paper attempts to address the fundamental barriers of learned optimization. The authors identify three barriers: computational requirements, number of training tasks and lack of inductive bias. A “large-scale” evaluation and comparison of learned optimizers is then carried out using many (1024) multi-core CPUs. A simple modification of an existing learned optimizer is also proposed that involves adding more input features. Unfortunately the results don’t seem to reveal any new insights.

Strengths
- The paper proposes a new, simple hierarchical learned optimizer that outperforms existing learned optimizers. The proposed model is very simple in theory but the implementation seems to still require quite a bit of “hand-engineering” in terms of selecting features etc.


- The experimental investigation reveals some interesting (albeit unsurprising) insights of large scale training of learned optimizers. These include things like training with more tasks improves performance, that learned optimization performs well in the hyper-parameter regime in which it was trained, that it learns some form of regularization and that it outperforms Adam when a non-optimal learning rate is used.


Concerns:
- My main concern with the paper is that some claims are over-blown. Although it is not clear at all that the current generation of learned optimizers can outperform hand-crafted optimizers, the paper makes misleading claims that can easily be taken out of context. Statements like, “We see this final accomplishment as being analogous to the first time a compiler is complete enough that it can be used to compile itself.” and “we believe learned algorithms will transform how we train models" are too strong given the current evidence of the performance of the learned optimizers. I would suggest the authors tone down these claims.


- The proposed hierarchical learned optimizer (as well as existing ones) seem to be more fragile than hand-crafted approaches such as Adam. For example, on CIFAR-10 in Figure 5 the learned optimizer fails even for batch sizes in the training regime. Is there any reason why this might be the case, especially considering that it has access to all the same information as Adam and Adam’s “hand-crafted” operations are quite simple?


- The disadvantages of the proposed learned optimizer still seems to outweigh the benefits. For example, the “careful tuning of learning rate schedules and momentum timescales” is traded instead for the selection of design of a sufficient range of tasks on which to train the optimizer. This seems to be a far more difficult task than just tuning a few hyperparameters. In addition, although hand-crafted optimizers “do not leverage alternative sources of information beyond the gradient”, the learned optimizers do not do much better and just seem to learn very simple regularization strategies. Currently the discussion on the advantages and disadvantages is completely separate. I think these need to be contrasted and compared on the grounds of what properties a user would prefer in an optimizer.


- The contribution of the paper in terms of new insight or knowledge is not clear. The “large-scale” training on a wide range of tasks and many unrolled steps is interesting but I’m not sure what new insights can be inferred from this? Furthermore, the hierarchical optimizer seems to be a small improvement of the mode proposed in (Wichrowska et al., 2017) with some additional input information (like validation loss).

---

> ### Author Response · Authors · 2020-11-19
> **Response**
>
> Thank you for your thoughtful review.
>
> \>  it is not clear at all that the current generation of learned optimizers can outperform hand-crafted optimizers
>
> We would like to emphasize that we performed a much more thorough experimental comparison against existing hand designed optimizers than has been done in any previous paper on learned optimizers. For instance, the baselines in Figure 3 allocate as much as 1000-2000x the compute to hyperparameter-tuning of hand designed optimizers for each task, compared to that used by the learned optimizer.
>
> The result of that thorough comparison is that … for some problems our learned optimizer outperforms hand designed optimizers which are exhaustively hyper-parameter tuned on the target problem … but for most problems they do not. Our learned optimizer does however typically outperform hand designed optimizers when only modest resources are spent tuning the hyperparameters of the hand designed optimizer, or when the hand designed optimizer is asked to perform well across many tasks with a single hyperparameter setting.
>
> We hope that we do not come across as making stronger statements than this. If we do, we are eager to add caveats and weaken them. We do stand by the performance claims expressed in the last paragraph.
>
> \> Statements like, “We see this final accomplishment as being analogous to the first time a compiler is complete enough that it can be used to compile itself.” and “we believe learned algorithms will transform how we train models" are too strong given the current evidence of the performance of the learned optimizers. I would suggest the authors tone down these claims.
>
> We have removed both of these sentences based on your feedback. Please let us know if there are other specific instances where you feel we overclaim.
>
> \> The proposed hierarchical learned optimizer (as well as existing ones) seem to be more fragile than hand-crafted approaches such as Adam ... Is there any reason why this might be the case, especially considering that it has access to all the same information as Adam and Adam’s “hand-crafted” operations are quite simple?
>
> This is an excellent observation, and one of the primary challenges facing the tiny subfield of learned optimizer research. We believe that Adam is more robust precisely *because* it is quite simple. Due to its simplicity, Adam is less expressive, and the best performance achievable by Adam (especially with fixed hyperparameters) will be worse than that achievable by a learned optimizer. On the other hand, because Adam is so simple it has a very limited ability to overfit to a specific optimization task, and so will tend to do better on out-of-distribution tasks.
>
> We do want to emphasize that despite the specific failure cases you highlight, in most cases our learned optimizer outperforms Adam-8p (Adam with additional hyperparameters for things like learning rate decay) with a fixed hyperparameter configuration.
>
>
> \> Advantages vs disadvantages of learned optimizers
>
> Our vision is to train a single optimizer that can be applied broadly without any form of per-task tuning. It is true that this comes at the cost of an expensive meta-training procedure, but this will only need to be performed once--in contrast with the per-task hyperparameter tuning that practitioners typically perform with a hand designed optimizer like Adam. Once this is done the learned optimizer can be applied to a new problem in a one shot fashion. In the case of standard hyper parameter tuning a lot of compute is typically employed to tune hyperparameters on any new problem. Throughout our work we strived show both advantages and disadvantages including in our experimental section (4.2, 4.4, 4.5, 4.6) as well as in the discussion section.
>
> \> Contributions -- no barriers overcome:
>
> We would like to reemphasize that this work is the first example of a learned optimizer architecture that performs competitively with aggressively hyper-parameter-tuned hand designed architectures.  To achieve this, we leverage a novel optimizer architecture, and a significantly larger distribution of tasks than previous work (Figure 2a).  While we agree with the reviewer's assessment that these ingredients do not represent a transformative qualitative shift in how to perform learned optimizer research, we argue that steady progress along this axis (i.e., careful architecture design coupled with large scale task distributions) *will* result in learned optimizers that unambiguously exceed hand designed optimizers in performance.

---

### Official Review · AnonReviewer3 · 2020-10-29
**Lot of promising experiments for learned optimizers but little insights**

**Rating:** 5
**Confidence:** 2

**Review:**

The authors propose to use a combination of Andrychowicz et al. LSTM based approach and Metz et.al feed forward network to learn an optimizer that is useful accross any task. The authors propose to use a few thousand tasks as developed in Metz et. al. for the purpose of training the optimizer. They focus on evolutionary strategies (ES) in lieu of unrolled optimization to keep the problem tractable.

The task is a challenging one and the authors propose a hierarchical optimizer similar to Wichrowska et. al. I wish the authors invested more thought into the architecture of the optimizer beyond assimilating features of previous work.

My main concern with the paper is lack of generalization capability of the proposed approach. The authors could have done a more thorough job evaluating the architecture by performing ablation studies, i.e., different permutation of inputs to the LSTM and feed-forward modules and understanding its effect. Fo e.g., how useful is the second moment or parameter values to FF; tensor shape/losses to the LSTM. I am also interested in understanding the trade-off involved in using ES over unrolled optimization. There should be some experiment discussing this trade-off as it seems to be a critical improvement over prior approaches. The sub optimal performance on imagenet is concerning.The authors should also provide guidance on task selection for optimizer training. This guidance could be coupled to the sub-par performance on imagenet.

---

> ### Author Response · Authors · 2020-11-19
> **Response**
>
> Thank you for your thoughtful review.
>
> \> My main concern with the paper is lack of generalization capability of the proposed approach.
>
> While better generalization is an ever-present goal, we would like to emphasize that:
> \- We do a much more thorough evaluation of generalization performance than *any previous or existing learned optimizer publication* (see Figures 3, 5, 6)
> \- We perform better than previous learned optimizers (Figure 2a).
> \- We often outperform hand designed optimizers on out of distribution tasks, even when the hand designed optimizer is given > 10x the compute resources with which to perform hyperparameter tuning on that task (Figure 3).
>
> \> I am also interested in understanding the trade-off involved in using ES over unrolled optimization. There should be some experiment discussing this trade-off as it seems to be a critical improvement over prior approaches.
>
> This was the focus of past work that we build on, providing experimental and theoretical exploration of this tradeoff: https://arxiv.org/abs/1810.10180. In short, unrolled optimization leads to extraordinarily high variance gradients and unstable outer training. This is primarily because the best performing optimizers tend to perform on the edge of stability, such that a very small change in the outer-parameters will cause inner-training to diverge. ES provides an unbiased estimate of the gradient of a smoothed version of the outer-loss, and the smoothed outer-loss does not have this extreme sensitivity to outer-parameter values. As a result, ES provides much lower variance gradient estimates than unrolled optimization. We have added additional discussion of this phenomenon to the manuscript.
>
> \> The sub optimal performance on imagenet is concerning.
>
> This is a misleading description of the performance. On ImageNet the learned optimizer performs similarly to aggressively learning-rate-tuned hand designed optimizers (Figure 6b), and does not require any tuning itself. We believe performing comparably to *tuned* hand designed optimizers, on a task that is different from those the optimizer was trained on, and without task-specific hyperparameter tuning, is a success.
>
> \> The authors should also provide guidance on task selection for optimizer training.
>
> We touch briefly on to align the training tasks with tasks we care about in section 3.2, and have also added a note in relation to our imagenet results as you suggest.

---

### Decision · Program_Chairs · 2021-01-07
**Final Decision**

**Decision:**

Reject

**Comment:**

The paper aims to address several challenges in learning neural network-based optimization algorithms by increasing the #unrolled steps, increasing the #training tasks, and exploring new parameterizations for the learning optimizer. The authors demonstrated the effectiveness of applying persisted Evolution Stratergies and backdrop through over 10,000 inner-loop steps can improve the performance of the learned optimizer. Empirical experiments showcased incorporating LSTM to the previous state-of-the-art improve their training performance.

There are a lot of interesting ideas in the paper. However, packaging them together and only glance over each idea briefly unfortunately dilutes the contribution and the novelty of the work. There are still some major concerns echoed among the reviewer:

1) The proposed hierarchical optimizer seems interesting. It is one of the major contributions of the paper. But, its architecture was only briefly mentioned in Sec 3.3. Its motivation, implementation and the corresponding engineering choices remain unclear by just reading the main text. Some of the details were discussed in the appendix but it would be of great interest if authors could give some intuition on which subset of the tasks the proposed architecture gives the most improvement / failure among the 6000 tasks.

2) Training the optimizer on a diverse set of tasks is crucial for the learned optimizer to generalize. One of the paper's contributions is to further expand the task dataset from the prior work Metz et al., (2020). The authors have conducted very thorough experiments on this new dataset, which is amazing. I would argue there are even enough results for another standalone paper. However, there is surprisingly little detail on how the newly proposed dataset differs from the prior TaskSet dataset. What are the new optimization problems? How are they different from the family of tasks in TaskSet? A TSNE plot of the tasks similar to Figure 1 from Metz et al. (2020) could provide more intuition for the reader and highlight the contribution.

Overall, if the authors could provide more insight into their experiments and the proposed methods, it would help the readers greatly to see the novelty and the contribution of the paper. The current version of the paper will need additional development and non-trivial modifications to be broadly appreciated by the community.